# Holistic Order Prediction in Natural Scenes

**Pierre Musacchio**
Seoul National University
pmusacchio@snu.ac.kr

**Hyunmin Lee**
LG AI Research
hyunmin@lgresearch.ai

**Jaesik Park**
Seoul National University
jaesik.park@snu.ac.kr

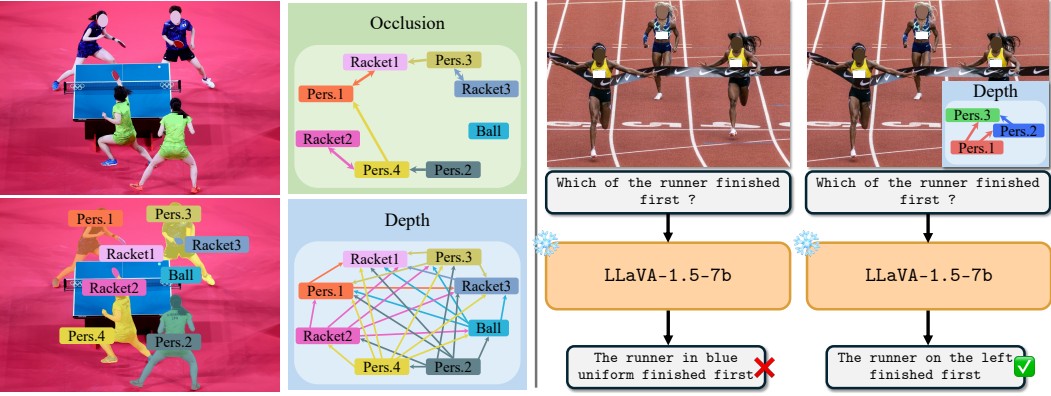

Figure 1: **In-the-wild inferences using InstaFormer[o,d]**. We display the segmentation results along the occlusion and depth order predictions of images gathered from the web. Instances are represented as nodes in the graphs. Edges are arrows characterizing their order relations. InstaFormer provides accurate instance-wise geometries. Moreover, converting its outputs to text format enables VLMs to understand geometries in a zero-shot manner better.

## Abstract

Even in controlled settings, understanding instance-wise geometries is a challenging task for a wide range of visual models. Although specialized systems exist, modern arts rely on expensive input formats (category labels, binary segmentation masks) and inference costs (a quadratic amount of forward passes). We mitigate these limitations by proposing **InstaFormer**, a network capable of **holistic order prediction**. That is, solely given an input RGB image, InstaFormer returns the full occlusion and depth orderings for all the instances in the scene in a single forward pass. At its core, InstaFormer relies on interactions between object queries and latent mask descriptors that semantically represent the same objects while carrying complementary information. We comprehensively benchmark and ablate our approach to highlight its effectiveness. Our code and models are open-source and available at this URL: https://github.com/SNU-VGILab/InstaOrder.

## 1 Introduction

While scene understanding has always been an active domain in computer vision [3, 19, 21, 24, 47, 50], its importance has recently become evident with the rising popularity of vision-language models (VLMs) [2, 5, 7, 27, 28]. In a VLM framework, the user textually interacts with an agent based on visual grounding from an input image [23, 27, 35, 42, 46]. However, current VLMs still struggle to understand the intricacies of the geometric relationships between the elements composing the scene. Instance-wise order prediction is often overlooked due to its apparent simplicity and to a

39th Conference on Neural Information Processing Systems (NeurIPS 2025).

tendency to opt for dense prediction networks [18, 34, 44, 45]. Yet, we emphasize that this simplicity is deceiving, even for dense foundation models [20, 45]. From determining pedestrian–car occlusions for the safe piloting of autonomous vehicles to estimating depth layouts and relative positioning in a robot's perception module, understanding instance-wise orderings is a *non-trivial* task that is often overlooked yet *crucial for grounding our interactions with machines*.

Few existing approaches explicitly tackle this geometrical scene graph problem [24, 47, 50]. Yet, they suffer from expensive inference costs and impractical input formats. Specifically, these methods formulate the occlusion and depth ordering prediction tasks a series of instance-wise edge prediction, meaning they require a quadratic amount of passes to obtain the full graph (Fig. 2a). Moreover, all the aforementioned arts require the binary masks of *all* the instances as input (Fig. 2b).

To address the input format issue, we first design a straightforward approach called Mask2Order. Mask2Order is a concatenation of a pre-trained segmentation network [4, 8, 9] to a pairwise order network [24]. Given an input image, the segmentation network first generates masks which are then fed by pairs, along with the input image to the pairwise network to obtain their geometrical relations. Although this approach alleviates the input limitation since it self-supplies the order head with generated masks, its performance degrades while continuing to require multiple inferences. Thus, we introduce the main contribution of our work, InstaFormer, a joint segmentation and ordering prediction network capable of *holistic* order prediction.

In the *holistic* prediction paradigm, we reformulate the edge-level occlusion and depth ordering prediction tasks to an adjacency matrix-level problem. This allows the network to predict the

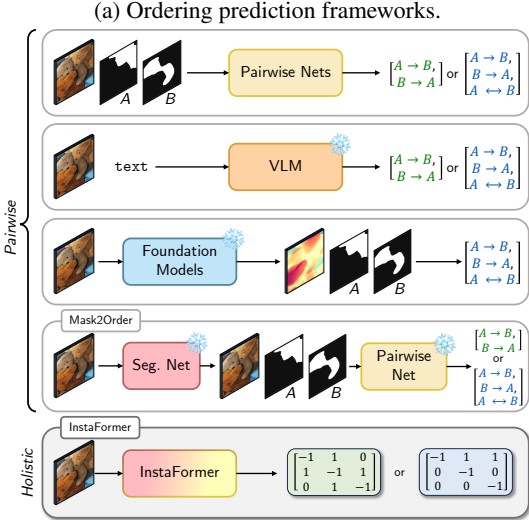

(a) Ordering prediction frameworks.

(b) Input-output and inference cost of ordering prediction frameworks.

| Methods | Input | | | Output ordering | |
|---|---|---|---|---|---|
| | Masks | Image | Text | Occlusion | Depth |
| Pairwise Nets | ✓ | ✓ | | ✓ | ✓ |
| VLM | | ✓ | ✓ | ✓ | ✓ |
| Foundation models | | ✓ | | | ✓ |
| Mask2Order | | ✓ | | ✓ | ✓ |
| **InstaFormer** | | ✓ | | ✓ | ✓ |

Figure 2: **Overview of our *holistic* approach**. We compare the holistic approach to other inference strategies (Fig. 2a) and highlight the input-output format discrepancies (Tab. 2b).

whole instance-wise geometries in a constant inference cost, *i.e.*, a single forward pass, for any arbitrary number of instances in the scene. This is made possible thanks to attention interactions between object queries and latent mask descriptors representing the same objects while carrying complementary information.

In summary, our contributions can be stated as follows:

- We cast the traditional edge-level occlusion and depth order problems to an adjacency matrix-level prediction task. We call this task *holistic* order prediction.
- We introduce **InstaFormer**, a network family capable of *holistic* order prediction, surpassing or matching the best available baselines on the tasks of occlusion and depth order prediction solely from an RGB input image.

## 2 Related Work

### 2.1 Order prediction

Occlusion order prediction has been introduced in the seminal work of [50] in which authors tackle the task of amodal instance segmentation. Amodal segmentation datasets such as COCOA [50] or KINS [32] subsequently annotate occlusion order annotations. Other datasets focus explicitly on labeling instance orderings, such as INSTAORDER [24] or WALT [37]. While PCNet-M is trained in an unsupervised manner [47], OrderNet [50] and InstaOrderNet [24] show superior performances by

being trained in a supervised setting. Inspired by instance-wise occlusion order, [24, 48] propose to also predict instance-wise *depth* orders. Ultimately, all these approaches leverage a pair of binary masks and an RGB image as inputs, making their real-world inference impractical. Additionally, the pairwise nature of these networks enforces a quadratic number of forward passes to obtain the full relations between all the instances in the scene.

On the other hand, we reformulate the pairwise edge-level ordering prediction problem to a single adjacency matrix-level prediction. This *holistic* property of our InstaFormer model yields the full geometrical ordering relations in a unique forward pass, solely from an RGB image given as input (Fig. 2a), effectively alleviating the input-output constraint induced by prior approaches while enabling single-pass inference.

## 2.2 Foundation models

Vision foundation models [20, 31, 44, 45] are trained on an extensive amount of annotations to excel in the task they were designed for. SAM can be prompted using points, bounding boxes, or masks to provide the most accurate segmentation corresponding to this prompt [20]. Similarly, Depth Anything models perform fine-grained monocular depth estimation [44, 45]. While initially designed to solve a specific task, their robustness and versatility allow them to stand as modules of a composite pipeline [38, 43]. In this art, we construct a simple baseline for order prediction by combining SAM and Depth Anything V2. Likewise, combining vision [33] and text foundation models [10, 40] together gives birth to a VLM [2, 27, 28]. Grounding the image [23, 35] in the textual latent space of the large language model enables performing diverse visual zero-shot prompting tasks directly using natural language queries.

To assess diverse types of models, we convert INSTAORDER to a visual-question answering (VQA) format named INSTAORDERVQA. Not only do we perform zero-shot prompting on LLaVA [27], but we also finetune it on INSTAORDERVQA to observe if VLMs are capable of understanding geometrical orderings. We also propose a simple foundation model pipeline consisting of SAM [20] and Depth Antyhing V2 [45].

# 3 Method

## 3.1 Problem formulation

Occlusion and depth orders are geometric relations linking instance nodes in a scene graph. Orthodox literature [24, 47] formulate these tasks as a series of edge inferences between pairs of instances $(A, B)$ living an input RGB image $\mathbf{I} \in \mathbb{R}^{H \times W \times 3}$ and their respective instance-level binary masks $(\mathbf{A}, \mathbf{B}) \in \{0, 1\}^{H \times W}$. Given $n \in \mathbb{N}$ such that $n \geq 2$ instances in $\mathbf{I}$, the occlusion (resp. depth) ordering task aims at predicting the $k$-categorical ordering matrix $\mathbf{G} \in \{0, ..., k\}^{n \times n}$ where each element $(i, j) \in \{0, \ldots, n-1\}^2$ represents the ordering relation, *i.e.* the edge, for instances $(i, j)$, namely $\mathbf{G}_{i,j}$. On the other hand, in our *holistic* scenario, we aim at bypassing edge-level inference of relations $\mathbf{G}_{i,j}$ and instead predict the adjacency matrix $\mathbf{G}$ all at once, directly from $\mathbf{I}$. Formally, we want to find a mapping $f$ such that $f(\mathbf{I}) = \mathbf{G}$.

From here, we use the following "→" notation to indicate both occlusion and depth relations. We read "$A \rightarrow B$" as "$A$ occludes $B$", for the occlusion order task and "$A$ is in front of $B$" for the depth order task. We refer the reader to Appendix A for more details about these tasks.

## 3.2 Preliminaries: pairwise networks

Existing arts [24, 47, 50] formulate the occlusion prediction outputs as a binary prediction problem, *i.e.*, $\{A{\rightarrow}B, B{\rightarrow}A\}$. This leads to no occlusion in the simplest case and a *bidirectional* occlusion in the most complex case. The depth ordering prediction task is formulated as a 3-classes classification problem $\{A{\rightarrow}B, B{\rightarrow}A, A{\longleftrightarrow}B\}$, with an extra prediction explicitly indicating if $A$ and $B$ share overlapping depth ranges [24] (Fig. 2a, pairwise networks).

All these approaches leverage input binary masks, making them cumbersome, especially when ground truth masks are unavailable at test time (Fig. 2a).

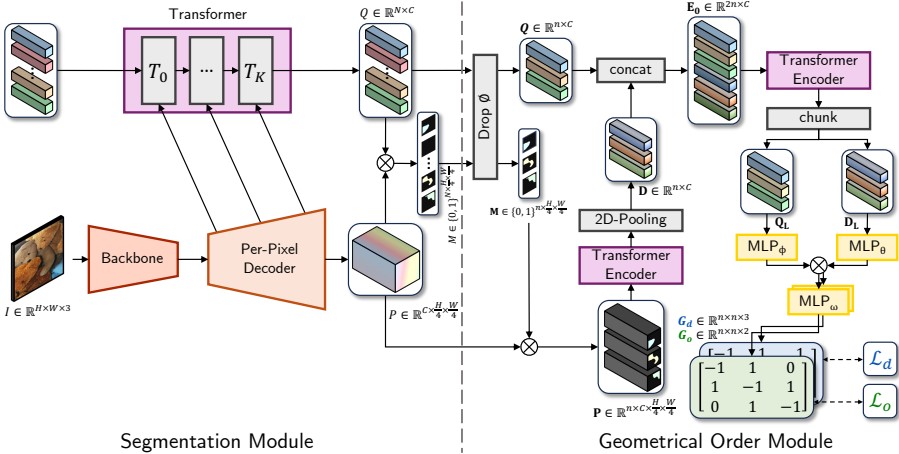

Figure 3: **Overview of InstaFormer**. The architecture comprises two modules. The first module generates the mask embeddings $Q$, the per-pixel embedding $P$, and the masks $M$. In practice, we use Mask2Former [9]. Then, a transformer-based geometrical order module predicts the orders from these three inputs. InstaFormer is capable of end-to-end *holistic* geometrical order predictions.

### 3.3 InstaFormer for holistic order prediction

**Motivations.** Here, we introduce our main contribution: the *InstaFormer* family of networks. InstaFormer is capable of *holistic* order prediction. Given an RGB image as input, it predicts the full order matrices in a single forward pass thanks to the *holistic* objective formulation and its internal attention mechanism use. The main idea of our method is to create *interaction between object queries and latent mask descriptors representing the same objects while carrying complementary information*. A dot product, akin to a similarity operator, computes the final geometrical output from these two representations. We invite the reader to follow equations along using Fig. 3.

**Architecture.** Let us assume a segmentation backbone $f$ designed using the meta-architecture as described by [9] and an input image $\mathbf{I} \in \mathbb{R}^{H \times W \times 3}$. Then, $f(\mathbf{I})$ returns the output mask embeddings $Q \in \mathbb{R}^{N \times C}$ from the transformer decoder and the per-pixel embedding output from the per-pixel decoder $P \in \mathbb{R}^{C \times \frac{H}{4} \times \frac{W}{4}}$. Here, $N$ denotes the number of object queries and $C$ the embedding dimension.

The binary masks $M \in \{0,1\}^{N \times \frac{H}{4} \times \frac{W}{4}}$ are obtained from $M = \lfloor (\sigma(QP) \rceil$, where the matrix multiplication can be seen as a dynamic convolution between a set of segment kernels and a feature map [17]. Resulting logits are sent to the sigmoid function $\sigma$ and binarized via the rounding function $\lfloor \cdot \rceil$ to obtain the final mask predictions.

Following DETR [4], we obtain the optimal assignment between ground truth and predicted segments using the Hungarian Matcher [22]. This way, we identify all the *no object* segments ($\emptyset$) and discard them, resulting in $n$ selected tokens $\mathbf{M} \in \{0,1\}^{n \times \frac{H}{4} \times \frac{W}{4}}$, $\mathbf{Q} \in \mathbb{R}^{n \times C}$. We now compute a per-mask-per-pixel embedding by masking $P$ using $\mathbf{M}$, resulting in a $\mathbf{P} \in \mathbb{R}^{n \times C \times \frac{H}{4} \times \frac{W}{4}}$ representation. This retains the values of the per-pixel embedding that falls in the region of the given segment while zeroing all others.

We want to obtain a global descriptor from each $\mathbf{p}_i \in \mathbf{P}$, with $i \in \{0, \dots, n\}$. We found that directly pooling from $\mathbf{P}$ does not provide a comprehensive enough description (Tab. 2b). Thus, we first forward $\mathbf{P}$ to a single transformer layer [11] and max-pool this updated representation instead. To obtain accurate segment-level mask descriptors, we restrict the attention of each $\mathbf{p}_i$ to itself using Masked Self-Attention (MSA):

$$\text{MSA}(\mathbf{P}, \mathcal{M}) = \text{softmax}\left(\frac{\mathbf{P}^Q \left(\mathbf{P}^K\right)^T + \mathcal{M}}{\sqrt{d}}\right)\mathbf{P}^V, \tag{1}$$

$$\text{where} \quad \mathcal{M} = \begin{cases} -\infty & \text{if} \quad \mathbf{p}_i[h,w] = 0, \\ 0 & \text{otherwise}. \end{cases} \tag{2}$$

Here, $\mathbf{P}^{\{Q,K,V\}} \in \mathbb{R}^{C \times d}$ are the respective queries, keys and values of $\mathbf{P}$ [41]. We only now apply max-pooling on the *masked* region of $\mathbf{P}$ to obtain our global descriptors:

$$\mathbf{D} = \text{maxpool}(\mathbf{P}) \in \mathbb{R}^{n \times C}. \tag{3}$$

While $\mathbf{D}$ contains spatially-grounded information, $\mathbf{Q}$ contains geometric and shape-relevant knowledge [17]. We concatenate the global descriptors $\mathbf{D}$ and the mask embeddings $\mathbf{Q}$ together, obtaining $\mathbf{E}_0 \in \mathbb{R}^{2n \times C}$ and forward this representation to $\{l\}_{l=1}^{L \in \mathbb{N}}$ transformer layers [11, 41] to generate interactions between those two token types [25]. We denote the final output of this module as $\mathbf{E}_L \in \mathbb{R}^{2n \times C}$.

We recover the updated mask embeddings $\mathbf{Q}_L$ and the descriptor tokens $\mathbf{D}_L$ by chunking $\mathbf{E}_L$ and then forward them to two different multi-layer perceptrons: $\mathbf{Q}^\star = \text{MLP}_\phi(\mathbf{Q}_L)$ and $\mathbf{D}^\star = \text{MLP}_\theta(\mathbf{D}_L)$.

We obtain the final geometric ordering prediction $\mathbf{G} \in \mathbb{R}^{n \times n}$ through matrix multiplication between the two representations: $\mathbf{G} = \mathbf{Q}^\star \mathbf{D}^{\star T}$, effectively computing the compatibility between each kernel and each global descriptor (Fig. 3). We project this representation one last time using task-specific MLPs with parameters $\omega$ for occlusion and $\delta$ for depth orderings, yielding the final occlusion order matrix $\mathbf{G}^o \in \mathbb{R}^{n \times n}$ and depth order matrix $\mathbf{G}^d \in \mathbb{R}^{n \times n}$ where the $(i, j)$-th element of the matrices represents the predicted geometric order between the $i$-th and $j$-th segments. When training InstaFormer$^o$ or InstaFormer$^d$, only one of these MLPs is initialized, whereas we use the two different MLPs simultaneously when training InstaFormer$^{o,d}$.

**Adapters.** To allow the network to fully benefit from the segmentation latent space of the mask extractor, we attach adapters to every feed-forward network (FFN) in the transformer decoder of the backbone [6, 16, 19]. We refer the reader to Appendix D for more information.

**Inference.** We cannot leverage the Hungarian Matcher at inference time since it requires ground truth annotations. Instead, we simply follow [9] and consider segments as detected if their confidence score lies above 0.8.

## 4 Experiments

### 4.1 Baselines

**VLM.** We introduce a VLM baseline by converting INSTAORDER to INSTAORDER-VQA and prompt the model with occlusion and depth order questions. We use LLaVA [27] as it is open-source, popular, and widely used. We run zero-shot evaluation but also fine-tune on our data (denoted with ‡ in Tab. 1). We refer the reader to Appendix B and C for more information.

**Non-parametric methods.** We employ the Y-axis and Area as two non-parametric methods to estimate occlusion and depth orders. These heuristics rely on the fact that an object at the bottom of the image might be closer to the camera (Y-axis), and objects that are in the foreground are usually bigger than objects in the background (Area).

**Foundation models.** By composing Grounded SAM [38] and Depth Anything V2 [45], we design a foundation pipeline capable of instance-wise depth order prediction. We refer the reader to Appendix C for more details.

**Pairwise paradigm.** Pairwise networks represent the most competitive baselines and provide the closest comparison to our setting. To the best of our knowledge, only three concurrent works attempt to predict geometrical orderings. Thus, we include them all. That is, we compare our work to PCNet-M [47], OrderNet [50], and InstaOrderNets [24].

**Mask2Order.** Since pairwise networks cannot perform inference unless a pair of segmentation masks is provided *a priori* [24, 47, 50] (Fig. 2a, pairwise networks), we introduce Mask2Order to solve this issue in the simplest possible way. It results from the concatenation of a segmentation backbone and a pairwise ordering prediction network (Fig. 2a, Mask2Order). Motivated by recent advances in this domain [4, 8, 49], we select Mask2Former [9] as our mask generator. While the

Mask2Order framework can obtain geometrical orderings from an RGB image without additional binary masks from the user, this network family still requires multiple forward passes to get the relations between all the instances. We also train InstaOrderNets on top of Mask2Former (denoted with ‡ in Tab. 1). In this setting, we use the same recipe as [24] but modernize optimization by switching the optimizer for AwamW and using a cosine schedule.

## 4.2 Datasets

We run experiments on INSTAORDER [24]. We convert INSTAORDER to a VQA version, *i.e.*, INSTAORDER-VQA, to evaluate LLaVA [28] on occlusion and depth order prediction in zero-shot and finetuned manners (denoted with ‡ in Tab. 1). We release the conversion script and the dataset with this work. We refer the reader to Appendix B for more information.

## 4.3 Evaluation protocol

**Metrics.** Following previous works [24, 47, 50], we evaluate the performance of our networks on the task of occlusion ordering prediction by reporting *precision*, *recall*, and *F1-score*. We compare each element of the predicted ordering matrix to each element of the ground truth matrix. Given two instances $A$ and $B$, the aforementioned metrics are expressed as follows:

$$\text{Recall} = \frac{\sum_{AB} \mathbb{1}(\hat{o}_{AB} = 1 \text{ and } o_{AB} = 1)}{\sum_{AB} \mathbb{1}(o_{AB} = 1)}, \tag{4}$$

$$\text{Precision} = \frac{\sum_{AB} \mathbb{1}(\hat{o}_{AB} = 1 \text{ and } o_{AB} = 1)}{\sum_{AB} \mathbb{1}(\hat{o}_{AB} = 1)}, \tag{5}$$

$$\text{F1-score} = \frac{2 \times \text{Precision} \times \text{Recall}}{\text{Precision} + \text{Recall}}, \tag{6}$$

where $o$ and $\hat{o}$ denote respectively ground truth and predicted occlusion order, and $\mathbb{1}$ is the indicator function.

We report the performances of our approaches for depth order prediction using Weighted Human Disagreement Rate (WHDR) [1]. WHDR indicates the percentage of weighted disagreement between ground truth $d$ and predicted depth order $\hat{d}$ [24]. Each annotation of INSTAORDER is accompanied by a weighting factor $w$ indicating the difficulty of the annotation based on how many annotation attempts it took for annotators to agree on the ground truth. WHDR is computed independently for every {distinct, overlap, all} category and is defined as follows:

$$\text{WHDR} = \frac{\sum_{AB} w_{AB} \cdot \mathbb{1}(\hat{d}_{AB} \neq d_{AB})}{\sum_{AB} w_{AB}}, \tag{7}$$

where $w_{AB} = \frac{2}{\text{count}_{AB}}$.

**Decoupling order evaluation from segmentation predictions.** In Mask2Order and InstaFormer, the geometrical prediction is performed after generating the segmentation masks. Thus, the performance of the ordering predictions is dependent on the segmentation predictions of the backbone model. To curb this dependency, we leverage the Hungarian Matcher [4] at evaluation time to provide a perfect matching between the predictions and the ground truths before performing the geometrical ordering evaluation. This way, we ensure that we obtain the full instance-wise evaluation for all the samples in the dataset, providing a fair assessment and allowing us to compare Mask2Order and InstaFormer with other baselines.

## 4.4 Evaluating naïve baselines

**Non-parametric methods.** We use the center of the masks as the Y-score and rank them accordingly to obtain the depth orders (Tab. 1, Y-axis). For the area method, we rank the masks by their size, the bigger the closer (Tab. 1, Area). These simple heuristics already provide a fair approximation of the geometric order of the elements in the scene.

Table 1: **Occlusion and depth order evaluation on INSTAORDER and INSTAORDER-VQA.** We benchmark LLaVA, non-parametric methods, pairwise approaches, foundation models, our Mask2Order baseline (zero-shot and trained, denoted with ‡), and our proposed *holistic* InstaFormer. All models are trained on INSTAORDER except for our foundation pipeline (off-the-shelf) and LLaVA (off-the-shelf and finetuned, denoted with ‡). Best results for a model family are in  yellow , best overall results are in **bold**.

| Dataset | Method | Input | | | | Output | | | Occlusion acc. ↑ | | | WHDR ↓ | | |
|---|---|---|---|---|---|---|---|---|---|---|---|---|---|---|
| | | GT Masks | Image | Category | Text | Occ. order | Depth order | Seg. | Recall | Prec. | F1 | Distinct | Overlap | All |
| VQA | LLaVA [27] | – | ✓ | – | ✓ | ✓ | ✓ | – | 49.98 | 48.28 | 34.10 | 37.37 | 44.56 | 39.52 |
| | LLaVA‡ [27] | – | ✓ | – | ✓ | ✓ | ✓ | – | 85.41 | 55.10 | 60.41 | 15.95 | 27.87 | 25.98 |
| INSTAORDER [24] | Area | ✓ | – | – | – | ✓ | – | – | 56.33 | 71.55 | 59.67 | 30.90 | 35.66 | 32.19 |
| | Y-axis | ✓ | – | – | – | ✓ | – | – | 44.84 | 57.34 | 47.30 | 22.19 | 39.04 | 29.20 |
| | PCNet-M [47] | ✓ | ✓ | – | – | ✓ | – | – | 59.19 | 76.42 | 63.02 | – | – | – |
| | OrderNet$^{M+I}$(ext.) [32] | ✓ | ✓ | – | – | ✓ | – | – | 84.93 | 78.21 | 77.51 | – | – | – |
| | InstaOrderNet$^o$(M) | ✓ | – | – | – | ✓ | – | – | 87.35 | 79.07 | 78.98 | – | – | – |
| | InstaOrderNet$^o$(MC) | ✓ | – | ✓ | – | ✓ | – | – | 88.70 | 78.21 | 79.18 | – | – | – |
| | InstaOrderNet$^o$(MIC) | ✓ | ✓ | ✓ | – | ✓ | – | – | 89.38 | 79.00 | 79.98 | – | – | – |
| | InstaOrderNet$^o$ | ✓ | ✓ | – | – | ✓ | – | – | 89.39 | 79.83 | 80.65 | – | – | – |
| | InstaOrderNet$^d$(M) | ✓ | – | – | – | – | ✓ | – | – | – | – | 22.96 | 30.46 | 25.23 |
| | InstaOrderNet$^d$(MC) | ✓ | – | ✓ | – | – | ✓ | – | – | – | – | 23.19 | 28.56 | 36.45 |
| | InstaOrderNet$^d$(MIC) | ✓ | ✓ | ✓ | – | – | ✓ | – | – | – | – | 13.33 | 26.60 | 17.89 |
| | InstaOrderNet$^d$ | ✓ | ✓ | – | – | – | ✓ | – | – | – | – | 12.95 | 25.96 | 17.51 |
| | InstaOrderNet$^{o,d}$ [24] | ✓ | ✓ | – | – | ✓ | ✓ | – | 82.37 | **88.67** | 81.86 | 11.51 | 25.22 | 15.99 |
| | MiDaS(Mean) [34] | ✓ | ✓ | – | – | – | ✓ | – | – | – | – | 10.42 | 37.67 | 21.70 |
| | MiDaS(Median) [34] | ✓ | ✓ | – | – | – | ✓ | – | – | – | – | 10.31 | 36.08 | 20.92 |
| | Foundation (Min-Max) | – | ✓ | – | – | – | ✓ | ✓ | – | – | – | 21.71 | 42.89 | 29.85 |
| | Foundation (Mean) | – | ✓ | – | – | – | ✓ | ✓ | – | – | – | 10.70 | 39.22 | 22.46 |
| | Foundation (Median) | – | ✓ | – | – | – | ✓ | ✓ | – | – | – | 10.80 | 39.11 | 22.36 |
| | Mask2Order$^o$ | – | ✓ | – | – | ✓ | – | ✓ | 84.37 | 78.48 | 77.49 | – | – | – |
| | Mask2Order$^{o‡}$ | – | ✓ | – | – | ✓ | – | ✓ | 77.92 | 88.58 | 79.1 | – | – | – |
| | Mask2Order$^d$ | – | ✓ | – | – | – | ✓ | ✓ | – | – | – | 14.16 | 27.15 | 18.52 |
| | Mask2Order$^{d‡}$ | – | ✓ | – | – | – | ✓ | ✓ | – | – | – | 12.96 | 28.6 | 18.40 |
| | Mask2Order$^{o,d}$ | – | ✓ | – | – | ✓ | ✓ | ✓ | 77.51 | 85.50 | 77.17 | 12.29 | 27.03 | 17.09 |
| | Mask2Order$^{o,d‡}$ | – | ✓ | – | – | ✓ | ✓ | ✓ | 79.81 | 86.86 | 79.09 | 12.44 | 28.44 | 18.19 |
| | InstaFormer$^o$ | – | ✓ | – | – | ✓ | – | ✓ | **89.82** | 78.10 | **81.89** | – | – | – |
| | InstaFormer$^d$ | – | ✓ | – | – | – | ✓ | ✓ | – | – | – | 8.47 | 24.91 | 13.73 |
| | InstaFormer$^{o,d}$ | – | ✓ | – | – | ✓ | ✓ | ✓ | 89.57 | 78.07 | 81.37 | **7.90** | **24.68** | **13.30** |

**VLM.** Zero-shot prompting LLaVA [27] scores the least out of all the benchmarked methods (Tab. 1, LLaVA). We hypothesize that this is because the pre-training of its visual encoder does not enforce precise spatial understanding [33]. We observe notable improvements under the finetuned setting (denoted with ‡), yet, performances remain behind visual experts.

**Foundation models.** While being incapable of performing occlusion order prediction, we find foundation models to obtain modest results: their performance still lag behind those of specialized networks even the ones that use the same format (Tab. 1).

**Mask2Order.** Table 1 confirms that Mask2Order performs worse than InstaOrderNet since it does not benefit from fine-grained segmentation input masks, even when InstaOrderNets are trained on top of Mask2Former (denoted with ‡). This implies that *predicting geometrical orderings from self-generated binary masks is a non-trivial task*. We further ablate the backbone size of Mask2Order in Tab. 4a and 4b (Appendix E). We observe a positive correlation between the size of the backbone and the accuracy results, which seem to confirm our hypothesis stating that pairwise networks are sensitive to the quality of the input mask.

## 4.5 Evaluating InstaFormer

**Experimental settings.** For all experiments, we train on 4 NVIDIA RTX A6000 for 120,000 iterations using AdamW [30] with learning rate $10^{-5}$ and reduce it to $10^{-6}$ and $10^{-7}$ at iterations 80,000 and 110,000 respectively, as suggested by [15]. We use a batch size of 16 and use the BCE an CE losses for occlusion and depth order respectively. We resize the input image to $1024\times1024$ and use RandomFlip during training, following [9]. For evaluation, we follow the default R-CNN baseline,

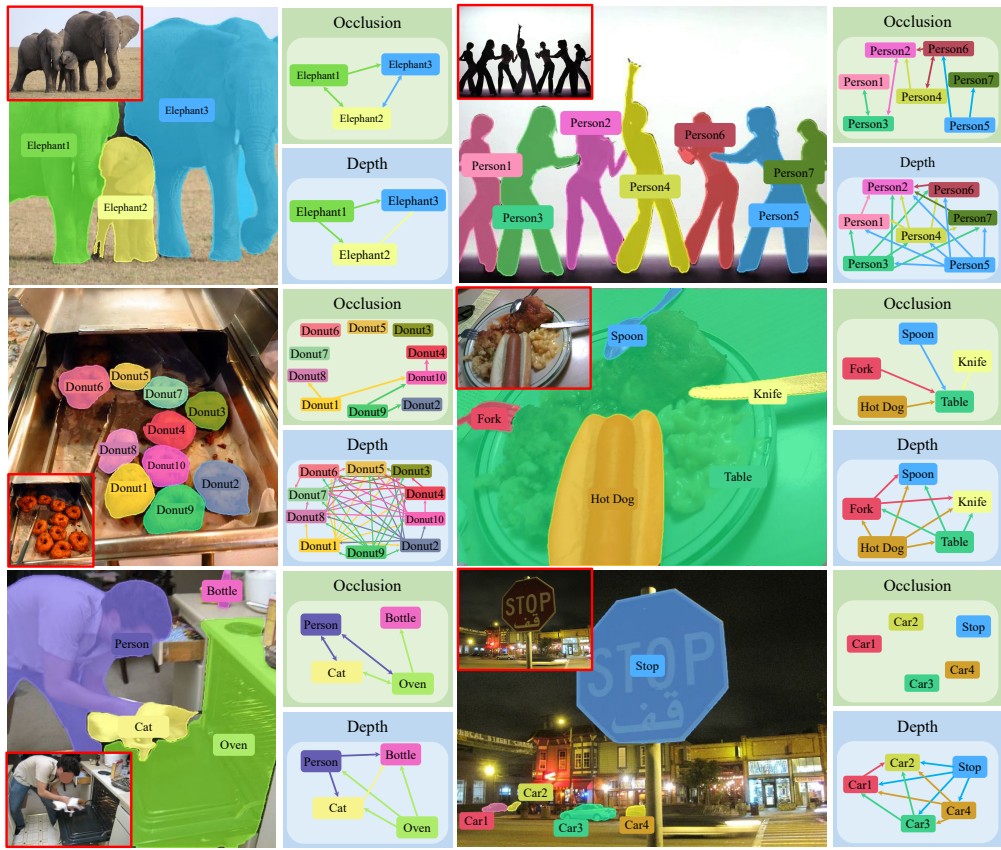

Figure 4: **Qualitative results obtained using our *holistic* InstaFormer[o,d] network**. The first row showcases in-the-wild examples from images gathered on the web. The remaining rows are extracted from the validation set of INSTAORDER. We show the segmentation predictions and represent their corresponding predicted occlusion and depth matrices as ordering graphs juxtaposed to them.

which consists of resizing the smaller size of the image to 800 pixels and the longer one to 1333. We match the segmentation predictions to their respective ground truth segments using Hungarian matching [22] before evaluating depth and occlusion order recovery.

We use 8 heads, 512-dimensional linear projections for all the attention layers, and 2-layered FFNs with 2,048 hidden nodes in all the transformer layers. The encoder consists of a single transformer layer that simply creates a global descriptor for each mask. On the other hand, the decoder comprises eight transformer layers. We add auxiliary losses on all transformer layers of the transformer decoder. This results in a 34M parameter geometrical ordering predictor. We initialize the entire ordering module using Xavier initialization [12]. We set the dimension of all the adapters to 64 and initialize their weights with Kaiming uniform [13] and their biases with zero initialization following [6].

**Occlusion order recovery.** We evaluate InstaFormer[o] in Tab. 1. It achieves the best F1 score of the benchmark, even overcoming InstaOrderNet[o] and InstaOrderNet[o,d], which were both trained using a simpler objective and on fine-grained ground truth binary masks. We ablate the segmentation backbone size of our InstaFormer[o], while keeping the geometric predictor untouched (cf. Tab. 4a (Appendix E). The more parameters, the better the results. This seems to imply that the ordering prediction results of the geometric predictor depend on the embedding space of the segmentation backbone. In all settings, InstaFormer[o] achieves notable results in comparison to Mask2Order[o]. While *holistic* order prediction is a challenging task, methods can be developed to outperform specialized pairwise models while alleviating the need for input binary masks and reducing the number of forward passes to 1. Most impressively, InstaFormer[o] slightly outperforms InstaOrderNet[o,d], the strongest pairwise network using fine-grained ground truth masks as inputs.

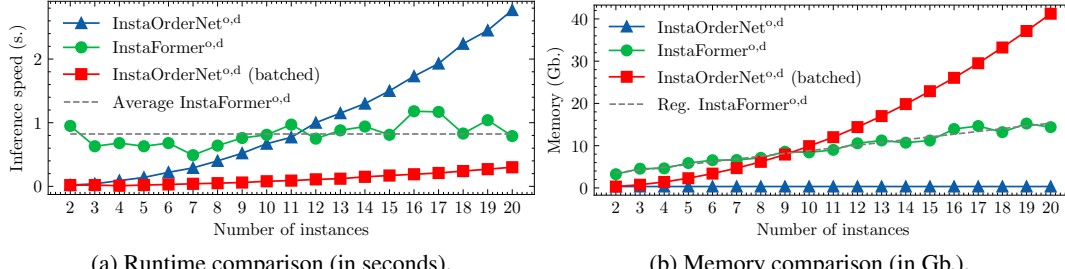

(a) Runtime comparison (in seconds).  (b) Memory comparison (in Gb.).

Figure 5: **Inference cost**. We benchmark the runtime and memory cost of InstaOrderNet[o,d] and our InstaFormer[o,d]. All measures are recorded on a single NVIDIA RTX A6000.

**Depth order recovery.** We present evaluation results of InstaFormer[d] in Tab. 1. InstaFormer[d] substantially outperforms all baselines on all WHDR (distinct, overlap, all). Similarly to InstaFormer[o], we also conduct an ablation study related to the backbone size of the segmentation network, while we keep the size of the geometric predictor fixed (Tab. 4b, Appendix E). Surprisingly, the backbone choice has little impact on the final results, which plateaus around the 13.70 mark. Contrary to occlusion order prediction, we hypothesize that depth order prediction does not require edge information and can still operate properly on rougher masks. Nonetheless, InstaFormer[d] exhibits remarkable results compared to InstaOrderNet[d] and Mask2Order[d], respectively being on par and beating them even though it operates in a much more challenging setting (*i.e.* holistic framework: without input masks and by predicting the full depth matrix at once).

**Joint occlusion and depth order recovery.** We report occlusion order and depth order evaluation for InstaFormer[o,d] in Tab. 1. InstaFormer[o,d] is competitive with all baselines and sometimes surpasses them, although not using fine-grained ground truth masks as input. It effectively ranks as the strongest depth order prediction network. Contrary to the Mask2Order family, joint occlusion and depth ordering prediction on the InstaFormer family performs similarly to order-specific networks. InstaFormer[o] marginally outperforms InstaFormer[o,d] (Tab. 1). We suspect the optimization process to be harder for joint occlusion and depth order prediction. In both cases, InstaFormer[o,d] obtains competitive results with InstaOrderNets, especially on depth prediction, even though it was not trained on fine-grained ground truth masks. Interestingly, slightly differently from what was reported in [24], we observe a substantial improvement in depth performances but not occlusion when performing joint occlusion and depth prediction. Thus, we still conclude that there are benefits to using a joint training scheme both from the computational efficiency and performance standpoints.

**Inference cost.** We benchmark the inference cost of our approach compared to InstaOrderNets, representative of pairwise approaches, in Fig. 5. InstaFormer compares favorably, as its runtime appears constant and memory linear with respect to the number of instances in the image. Details can be found in Appendix F.

### 4.6 Ablation studies

**Input modality.** The main idea behind our method is to generate instance-wise interactions between latent object representations. Naïve interaction from object queries to themselves results in low occlusion accu-

Table 2: **Ablation study**. Occlusion and depth order performances on INSTAORDER using InstaFormer[o,d] Swin-T.

(a) Ablation on the input modality.

| Input modality | Occlusion acc. ↑ | | | WHDR ↓ | | |
|---|---|---|---|---|---|---|
| | Precision | Recall | F1 | Distinct | Overlap | All |
| Queries/Queries | 73.56 | 88.45 | 77.92 | 15.85 | 39.83 | 23.21 |
| Descriptors/Descriptors | 75.68 | **88.97** | 79.39 | 16.75 | 38.11 | 22.56 |
| Queries/Descriptors | **88.64** | 75.56 | **79.74** | **8.43** | **25.36** | **14.03** |

(b) Ablation on the pooling strategy.

| Pooling | Transformer layers | Occlusion acc. ↑ | | | WHDR ↓ | | |
|---|---|---|---|---|---|---|---|
| | | Precision | Recall | F1 | Distinct | Overlap | All |
| Max | 0 | 74.86 | **87.87** | 79.01 | 16.08 | 37.87 | 21.88 |
| Max | 1 | **88.64** | 75.56 | 79.74 | **8.43** | **25.36** | **14.03** |
| Max | 2 | 87.62 | 78.30 | 79.80 | 15.67 | 34.85 | 21.48 |
| Max | 3 | 87.39 | 78.25 | **79.81** | 15.67 | 37.72 | 21.57 |

racy and poor depth ordering performances (Tab. 2a). When latent mask descriptors interact with themselves, results for occlusion orders rise, but not for depth orders. Best performances are obtained when combining information from both object queries and latent mask descriptors, as they carry complementary information while referring to the same objects.

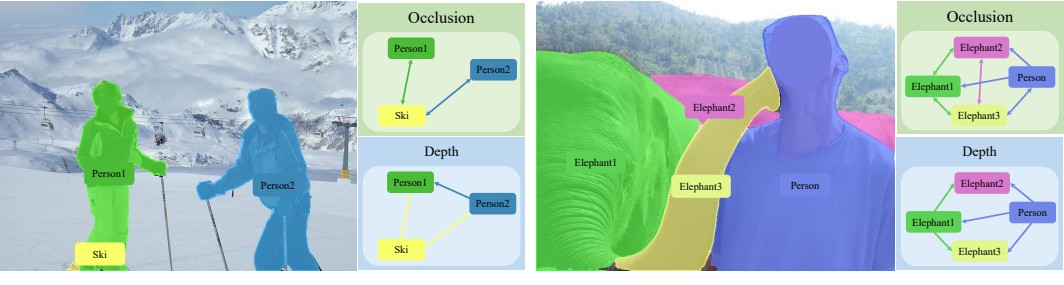

| (a) Instance-mixup. | (b) Segmentation failure. |

Figure 6: **Failure cases of InstaFormer[o,d]**. We highlight two failure cases of our network: the "instance-mixup" (Fig. 6a) and the "segmentation failure" (Fig. 6b).

**Pooling strategy.** Naïvely constructing the latent mask descriptor does not enhance the performances as much as it could in Tab. 2b. In fact, directly max-pooling the latent masks does not yield good performance in the depth order prediction task. Instead, simply adding a transformer layer before pooling produces a more comprehensive descriptor, enabling competitive results for occlusion and depth order prediction. As the performances saturate with a single layer, we do not scale more.

### 4.7 Qualitative results

InstaFormer can accurately determine geometrical orderings in complex scenes, even for small objects (donuts, cars in Fig. 4) or under in-the-wild inference (K-pop in Fig. 4, table tennis in Fig. 1). Our network still predicts reasonable relations even under sparse clue inputs (K-Pop example). After text conversion, InstaFormer's outputs can geometrically ground VLMs and enhance their reasoning in zero-shot prompting (Fig. 1, right). We provide diverse qualitative results of our model on INSTAORDER in Fig. 10, in Appendix G. We observe that our model predicts sane orderings in various scene types. Ground truths for all the predictions are available in Fig. 9 and Fig. 11 in the Appendices.

### 4.8 Failure cases

We highlight two cases in which InstaFormer struggles to predict accurate relationships between instances Fig. 6. We name these situation "instance mixup" and "segmentation failure". In the "instance-mixup" case, the "Person" instances related to the "Ski" are shared instead of being targeted solely on "Person1". In the "segmentation failure" case, a singular instance "Elephant" is wrongfully segmented multiple times, leading to multiple false positive nodes in the geometrical graphs.

## 5 Conclusion

We benchmark a diverse range of model types from foundation models to VLM, passing through visual experts on the tasks of instance-wise occlusion and depth order prediction. We propose InstaFormer to loosen the input-output constraints of existing methods to a bare RGB image while reducing the inference cost of the network to a single forward pass thanks to its *holistic* nature. InstaFormer matches or surpasses all baselines on the tasks of interest.

**Limitations and future work.** INSTAORDER annotations are limited to 10 instances per image. It would be interesting to ensure the generality of our approach by obtaining GT annotations on a larger number of objects. We also plan to extend our work to VLMs and layer-aware generative modeling.

## Acknowledgments

This work was supported by the IITP grant (RS-2021-II211343: AI Graduate School Program at Seoul National University (5%), RS-2024-00509257: Global AI Frontier Lab (40%), and RS-2025-25442338: AI Star Fellowship Support Program (55%)) funded by the Korean government (MSIT).

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

# Appendix

## A Task definition

Since the tasks of occlusion and depth order prediction are not commonly tackled in the computer vision community, we describe the format of these two challenges in details.

Let $\mathbf{I} \in \mathbb{R}^{H \times W \times 3}$ be an image. Given that the image contains $n \in \mathbb{N}$ instances, we first discriminate the cases where $n = 0$ and $n = 1$. When $n = 0$, there cannot be any geometrical relations since there are no instances. When $n = 1$ the geometrical relations between the only instance in the image and itself is undefined. Thus, it only makes sense of trying to compute occlusion and depth orderings in the event that $n \geq 2$. Hence, suppose it is the case.

Occlusion and depth orders are to be understood as relations in a graph that can itself be represented as an adjacency matrix $\mathbf{G} \in \{0, \dots, k\}^{n \times n}$. Here, each entry correspond to a possible relation type. Generally, we assume $k \in \mathbb{N}$ possible relations.

Specifically, The occlusion order matrix $\mathbf{G}^o$ contains $k = 2$ possible valid entries. Namely, for a pair $(i, j) \in [\![2, \dots, n]\!]^2$:

- $\mathbf{G}^o_{i,j} = 0$ indicates that instance $i$ does not occlude instance $j$,
- $\mathbf{G}^o_{i,j} = 1$ indicates that instance $i$ occludes instance $j$.

We call the case $\mathbf{G}^o_{i,j} = \mathbf{G}^o_{j,i} = 1$ a *bidirectional* occlusion.

Similarly, the depth order matrix $\mathbf{G}^d$ contains $k = 3$ possible valid entries. Namely, for a pair $(i, j) \in [\![2, \dots, n]\!]^2$:

- $\mathbf{G}^d_{i,j} = 0$ indicates that instance $i$ is not in front of instance $j$,
- $\mathbf{G}^d_{i,j} = 1$ indicates that instance $i$ is in front of instance $j$,
- $\mathbf{G}^d_{i,j} = 2$ indicates that instances $i$ and $j$ share similar overlapping depth ranges.

The upper triangle of the depth order matrix conditions the lower part of the matrix since if $\mathbf{G}^d_{i,j} = 1$, then $\mathbf{G}^d_{j,i} = 0$ and vice versa. Additionally, if $\mathbf{G}^d_{i,j} = 2$, then $\mathbf{G}^d_{j,i} = 2$.

Note that in both the occlusion and depth order prediction task, computing $\mathbf{G}_{i,i}$ does not make sense, thus we simply manually fill these entries with $-1$.

Predicting occlusion and depth order relations thus consist in predicting each entry in $\mathbf{G}$. Traditional pairwise networks model this problem at the edge-level, meaning that a forward pass of the model predicts a single $\mathbf{G}_{i,j}$. Such paradigm enforces a quadratic constraint of forward passes with respect to the number of objects $n$ since the size of the matrix grows quadratically for increasing $n$'s.

On the other hand, we propose to reformulate this problem from and edge-level prediction task to an adjacency matrix-level prediction problem, meaning predicting the full matrix $\mathbf{G}$ in a single forward pass. We call this task *holistic* order prediction.

## B About INSTAORDER-VQA

To evaluate a wide range of models, we propose a baseline for occlusion and depth order prediction based on LLaVA [27, 28] (see Tab. 1 in the main paper). In order to evaluate LLaVA on INSTAORDER, we first need to convert the occlusion and depth matrices into a visual question answering (VQA) format.

### B.1 Visual question answering

We convert every pairwise annotation in INSTAORDER to a textual annotation ready to be fed into LLaVA. Specifically, given two instances $A$ and $B$ in an image, we ask LLaVA the following question:

- `Is the {A_cls} {REL} the {B_cls} ?  Answer the question in a single word.`

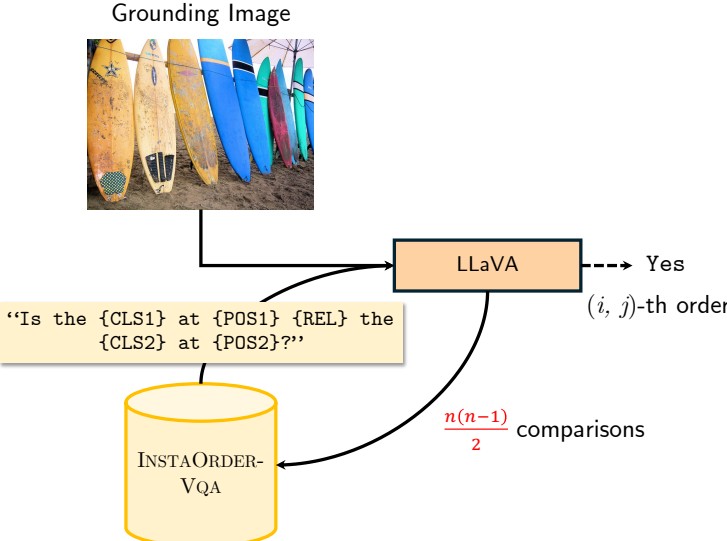

Figure 7: **Overview of our VLM pipeline**. We use LLaVA [27] to obtain geometrical orderings and evaluate data from our converted INSTAORDER-VQA dataset. We refer the reader to Appendix C.2 for more information.

Here, {REL} indicates the ordering relation, specifically:

- For occlusion orderings, {REL} = "obstructing",

- For depth orderings, {REL} = "closer to us than".

$\{A_{\mathrm{cls}}\}$ and $\{B_{\mathrm{cls}}\}$ respectively indicate the ground truth class of $A$ and the ground truth class of $B$. They are replaced dynamically for each question based on the currently compared instances. This strategy is sufficient to obtain a binary yes-no answer from LLaVA that can further be converted back to its respective element in the occlusion or depth matrix simply by mapping yes to 1 and no to 0.

### B.2 Instance-wise disambiguation

However, there still lies ambiguity when two instances in the scene share the same class since then $A_{\mathrm{cls}} = B_{\mathrm{cls}}$. Thus, it is unclear for the model to know which string refers to which instance in the image. To mitigate that issue, we leverage the fact that LLaVA can process bounding box coordinates and input the coordinates of the instances to clarify the position of the object we refer to (Fig. 7).

Therefore, following LLaVA [27, 28], we encode the bounding box of an ambiguous INSTAORDER instance as a 4-tuple $[a_w, a_h, b_w, b_h]$. Here, $a$ represents the top left corner while $b$ represents the bottom right corner of the bounding box. We then normalize the coordinates to fall in the $[0, 1]$ range. If objects can be identified without ambiguity, we do not specify their bounding boxes in the prompt.

Converting the annotations of the 4,071 images of the validation set of INSTAORDER to INSTAORDER-VQA yields a total of 178,539 VQA ordering prompts, along with their respective ground truths.

## C Baselines

### C.1 Benchmark specifications

We specify the backbones used for our main benchmark (Tab. 1) of the main paper in Tab. 3. Note that Mask2Order and InstaFormer's backbones are frozen at training time.

Table 3: **Backbone specifications for benchmarked methods**. We refer the reader to table 1 for the actual results.

| Method | Backbone |
|--------|----------|
| LLaVA [28] | CLIP [33] + Vicuna [10] |
| Area | – |
| Y-axis | – |
| PCNet-M [47] | U-Net [39] |
| OrderNet$_{M+I}$ (ext.) [32] | ResNet-50 [14] |
| InstaOrderNets [24] | ResNet-50 [14] |
| MiDaS [34] | ResNet-50 [14] |
| Foundation | DAv2 [45] + SAM [36] |
| Mask2Order | Mask2Former [9] |
| InstaFormer | Mask2Former [9] |

## C.2 LLaVA

**Benchmark.** For our benchmark (Tab. 1, in the main paper), we use LLaVA 1.5 [27], which is composed of a CLIP ViT-L 336 pixels vision encoder and a Vicuna 7B [10] token generator.

We provide two settings for this experiment. In the first setup, we simply use the off-the-shelf LLaVA model and perform zero-shot evaluation. Since the results are lower than expert baselines, we propose a stronger setting in which we finetune LLaVA onto INSTAORDERVQA.

In both cases, we prompt the model with the INSTAORDER-VQA evaluation set and report the results using the same metrics as with the other methods. A description of our pipeline is detailed in Fig. 7 of the supplementary material.

**Finetuning.** We observed that finetuning the model directly after the pre-training stage results in mode collapse, in which the VLM constantly outputs no for any ordering prompt. Instead, we take a visual instruction-tuned model and fine-tune it from there on our custom INSTAORDERVQA data.

We start by splitting the dataset into four categories: occlusion prompts to which the answer is Yes, occlusion prompts to which the answer is no, depth prompts to which the answer is yes and depth prompt to which the answer is No. We then randomly shuffle the samples in each category.

Since the number of parameters of LLaVA is order of magnitudes above InstaFormer, we opt not to train on the full dataset to obtain a more reasonable comparison. Specifically, we compute the ratio between the number of parameters of LLaVA and our model. We find that our model has 2% of the parameters contained in a LLaVA model. Thus, we subsample the training set of INSTAORDERVQA to 2% of the annotations. This means that we sample 58K annotations for training in total. We evenly select those annotations from the four categories mentioned in the previous paragraph to obtain a balanced dataset. We finetune LLaVA for a single epoch on 8 NVIDIA RTX A6000 with a batch size of 16 per GPU using the finetuning scripts from the official repository. We use all the default hyperparameters.

## C.3 Foundation models

**Foundation pipeline.** A depth order prediction can be viewed as a comparison between two instance-level depth maps. Considering this idea, we can compose foundation models together to obtain an instance-level depth estimator.

We use Grounded SAM [20, 38] and Depth Anything V2 [45] to obtain instance-level depth estimations that we can then compare together to obtain depth ordering predictions. A diagram of the overall pipeline is depicted in Fig. 8. To propose a baseline as competitive as possible, we select

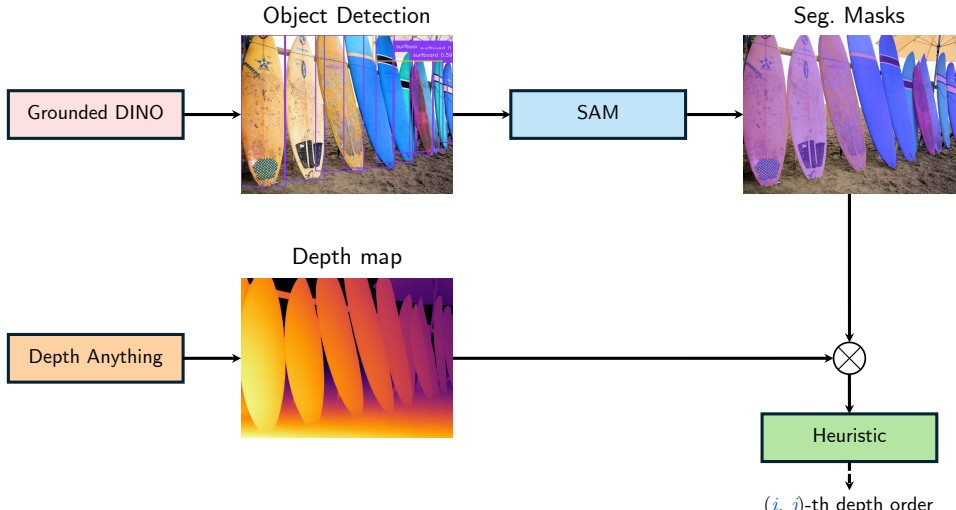

Figure 8: **Overview of our foundation model pipeline**. We create instance-level depth orderings from composing the predictions of Grounded SAM [38] and Depth Anything V2 [45]. We create the instance-wise depth predictions by masking the depth prediction using the instance segments generated from grounded SAM. Finally, we extract the prediction using a heuristic (min-max, mean, median). We refer the reader to Appendix C.3 for more details.

the best-performing networks available at the time of writing, *i.e.*, Grounding DINO Swin-T[1], SAM ViT-H[2] and Depth Anything V2 ViT-L[3].

We denote Grounded SAM [38] as $f_\theta$, a segmentation model $f$ parameterized by $\theta$ and Depth Anything V2 as $g_\phi$ a monocular depth estimation model $g$ parameterized by $\phi$. We start by listing all the instance classes of COCO [26], denoted $C = \{\texttt{cls}_1, \ldots, \texttt{cls}_K\}$ in a textual format and use them as text conditioning for Grounded SAM [38] along with the image of interest $\mathbf{I} \in \mathbb{R}^{H \times W \times 3}$:

$$\mathbf{S} = f_\theta(\mathbf{I}; C), \tag{8}$$

where $\mathbf{S} = \left\{ s_i \mid s_i \in \{0,1\}^{H \times W} \right\}_{i=0}^N$ are the binary segmentation masks for $N$ found instances in $\mathbf{I}$ that fall in any class of $C$.

We concurrently forward the image into the monocular depth predictor $g_\phi$:

$$\mathbf{D} = g_\phi(\mathbf{I}), \tag{9}$$

where $\mathbf{D} \in \mathbb{R}^{H \times W}$ represents the depth estimation of $\mathbf{I}$. We now obtain the segment-level depth maps by masking $\mathbf{D}$ using each $s_i$:

$$\mathbf{D}^{\text{order}} = \mathbf{D} \otimes \mathbf{S} = \{\mathbf{D} \odot s_i, \forall s_i \in \mathbf{S}\} \subset \{0,1\}^{N \times H \times W}, \tag{10}$$

where $\odot$ is the Hadamard product. We further derive the final depth orderings from $\mathbf{D}^{\text{order}}$ using a simple heuristic.

**Depth order heuristics.** There are several ways to obtain instance-wise depth orderings given segment-level depth maps. In our work and benchmark, we report three heuristics that yield different results (reported in Tab. 1, in the main paper): *min-max*, *mean*, and *median*.

The min-max heuristic revolves around extracting the minimum and maximum of each $\mathbf{D}_i^{\text{order}}$ to obtain a spanned depth range for every instance:

$$\begin{aligned} d_i^{\text{start}} &= \min(\mathbf{D}_i^{\text{order}}) \\ d_i^{\text{end}} &= \max(\mathbf{D}_i^{\text{order}}) \end{aligned} \quad, \quad \forall i \in \{1, \ldots, N\}. \tag{11}$$

---

[1]The Grounded DINO model for Grounded SAM is available at this URL.

[2]The SAM model zoo is available at this URL.

[3]The Depth Anything V2 model zoo is available at this URL.

Table 4: **Ablation study related to the backbone size**. We benchmark Mask2Order and InstaFormer for depth order prediction on INSTAORDER [24]. We experiment with various common sizes of Swin transformers [29]. We subscript the number of queries for every backbone size and specify IN-21K pre-trained backbones using †. The best results for a model family are in ⬛yellow⬛ , and the best overall results are in **bold**.

(a) Occlusion order prediction results.

| Method | Backbone | Occlusion acc. ↑ | | |
|---|---|---|---|---|
| | | Precision | Recall | F1 |
| Mask2Order$^o$ | Swin-T$_{100}$ | 83.23 | 77.96 | 76.74 |
| | Swin-S$_{100}$ | 83.08 | 78.17 | 76.66 |
| | Swin-B$_{100}$ | 83.62 | 78.06 | 76.89 |
| | Swin-B$^\dagger_{100}$ | 83.94 | 78.39 | 77.24 |
| | Swin-L$^\ddagger_{200}$ | 84.37 | 78.48 | 77.49 |
| Mask2Order$^{o,d}$ | Swin-T$_{100}$ | 76.20 | 84.46 | 76.11 |
| | Swin-S$_{100}$ | 76.23 | 85.15 | 76.36 |
| | Swin-B$_{100}$ | 77.03 | 84.91 | 76.78 |
| | Swin-B$^\dagger_{100}$ | 76.93 | 85.06 | 76.77 |
| | Swin-L$^\ddagger_{200}$ | 77.51 | **85.50** | 77.17 |
| InstaFormer$^o$ | Swin-T$_{100}$ | 89.06 | 75.69 | 79.63 |
| | Swin-S$_{100}$ | 88.91 | 77.31 | 80.53 |
| | Swin-B$_{100}$ | 89.02 | 76.95 | 80.64 |
| | Swin-B$^\dagger_{100}$ | 89.53 | 77.34 | 80.99 |
| | Swin-L$^\ddagger_{200}$ | 89.82 | 78.10 | **81.89** |
| InstaFormer$^{o,d}$ | Swin-T$_{100}$ | 88.64 | 75.56 | 79.74 |
| | Swin-S$_{100}$ | 88.20 | 75.98 | 79.57 |
| | Swin-B$_{100}$ | 88.47 | 75.96 | 79.72 |
| | Swin-B$^\dagger_{100}$ | 89.24 | 76.66 | 80.34 |
| | Swin-L$^\ddagger_{200}$ | 89.57 | 78.07 | 81.37 |

(b) Depth order prediction results.

| Method | Backbone | WHDR ↓ | | |
|---|---|---|---|---|
| | | Distinct | Overlap | All |
| Mask2Order$^d$ | Swin-T$_{100}$ | 14.73 | 27.48 | 19.02 |
| | Swin-S$_{100}$ | 14.48 | 27.66 | 18.81 |
| | Swin-B$_{100}$ | 14.31 | 27.73 | 18.75 |
| | Swin-B$^\dagger_{100}$ | 14.54 | 27.39 | 18.75 |
| | Swin-L$^\ddagger_{200}$ | 14.16 | 27.15 | 18.52 |
| Mask2Order$^{o,d}$ | Swin-T$_{100}$ | 12.96 | 27.43 | 17.78 |
| | Swin-S$_{100}$ | 12.67 | 27.62 | 17.60 |
| | Swin-B$_{100}$ | 12.49 | 27.29 | 17.35 |
| | Swin-B$^\dagger_{100}$ | 12.70 | 27.35 | 17.48 |
| | Swin-L$^\ddagger_{200}$ | 12.29 | 27.03 | 17.09 |
| InstaFormer$^d$ | Swin-T$_{100}$ | 8.10 | 25.43 | 13.75 |
| | Swin-S$_{100}$ | 8.44 | 26.04 | 14.48 |
| | Swin-B$_{100}$ | 8.28 | 25.05 | 13.88 |
| | Swin-B$^\dagger_{100}$ | 8.15 | 25.19 | 13.72 |
| | Swin-L$^\ddagger_{200}$ | 8.47 | 24.91 | 13.73 |
| InstaFormer$^{o,d}$ | Swin-T$_{100}$ | 8.43 | 25.36 | 14.03 |
| | Swin-S$_{100}$ | 8.54 | 25.42 | 13.96 |
| | Swin-B$_{100}$ | 8.84 | 25.77 | 14.39 |
| | Swin-B$^\dagger_{100}$ | 8.15 | 25.79 | 14.06 |
| | Swin-L$^\ddagger_{200}$ | **7.90** | **24.68** | **13.30** |

Subsequently, sorting the resulting ranges provides the desired depth orderings. In practice, the min-max heuristic is slightly flawed as the instance masks are not always perfectly aligned with the instance-level depth maps. This can result in obtaining minimum or maximum values slightly outside the actual instances, generating bias in the sorting process. Instead, we find the other two heuristics simpler and more robust to noise (as results reveal in Tab. 1 of the main paper).

The mean heuristic relies on obtaining the mean of each instance-level depth map:

$$d_i^{\text{mean}} = \text{mean}(\mathbf{D}_i^{\text{order}}), \quad \forall i \in \{1, \dots, N\}. \tag{12}$$

Then, using these scalar values as proxies for the distance of each mask from the camera, we perform a sorting to obtain the final instance-wise orderings.

The median heuristic works the same way as the mean heuristic, simply differing by using the median instead of the mean:

$$d_i^{\text{median}} = \text{median}(\mathbf{D}_i^{\text{order}}), \quad \forall i \in \{1, \dots, N\}. \tag{13}$$

In practice, we find the median heuristic to perform the best (cf. Tab. 1, in the main paper).

# D  Adapters

Since we use frozen Mask2Former networks for our experiments, we use adapters to improve the quality of the trainable geometrical ordering predictors of InstaFormer. Specifically, we attach adapters from the attention mechanism to the FFNs (feed forward networks) of every transformer layer in the segmentation transformer and pixel-decoder modules [6, 19].

We supervise the adapters using all losses from Mask2Former [9] as well as the losses of the geometrical module, *i.e.*, $\mathcal{L}_o$ for InstaFormer$^o$, $\mathcal{L}_d$ for InstaFormer$^d$ or $\mathcal{L}_o + \mathcal{L}_d$ for InstaFormer$^{o,d}$ and weight them by a factor $\lambda_o = \lambda_d = 5$.

We conduct experiments to study the effects of these components on InstaFormer in Appendix E.2.

Table 5: **Ablation study related to using adapters in the segmentation backbone of our InstaFormer networks**. We report the results on the occlusion and depth order prediction performances on INSTAORDER for the InstaFormer family of networks. The best results for a model family are in yellow , and the best overall results are in **bold**.

| Method | Backbone | Adapters | Occlusion acc. ↑ | | | WHDR ↓ | | |
|---|---|---|---|---|---|---|---|---|
| | | | Precision | Recall | F1 | Distinct | Overlap | All |
| InstaFormer$^o$ | Swin-L$_{200}^{\ddagger}$ | | 89.00 | 75.21 | 79.71 | – | – | – |
| | | ✓ | **89.82** | **78.10** | **81.89** | – | – | – |
| InstaFormer$^d$ | Swin-L$_{200}^{\ddagger}$ | | – | – | – | 17.10 | 39.37 | 23.52 |
| | | ✓ | – | – | – | 8.47 | 24.91 | 13.73 |
| InstaFormer$^{o,d}$ | Swin-L$_{200}^{\ddagger}$ | | 88.96 | 75.81 | 80.05 | 17.34 | 39.58 | 23.40 |
| | | ✓ | 89.57 | 78.07 | 81.37 | **7.90** | **24.68** | **13.30** |

# E    Ablations studies

## E.1    Segmentation backbone size

We perform an ablation study on the size of the segmentation backbone with respect to the performance for occlusion and depth ordering prediction for both Mask2Order and InstaFormer. In all experiments, the *backbones are frozen* and *we only train the geometrical predictor on top of them*.

**Occlusion order recovery.** We report the results when varying the backbone of the segmentation network from Swin-T$_{100}$ to Swin-L$_{200}^{\ddagger}$ in Tab. 4a. For Mask2Order$^o$, we observe a noticeable performance increase when using IN-21K pre-trained backbones. We observe a similar trend for Mask2Order$^{o,d}$.

This is similar to both InstaFormer$^o$ and InstaFormer$^{o,d}$, where the most noticeable performance improvement comes from increasing the queries to 200 and the backbone size to Swin-L, which in both cases results in a full point of improvement over their Swin-B$_{100}^{\ddagger}$ counterparts. While pre-training the backbones on ImageNet-21K for InstaFormer seems beneficial, it only yields half a point of F1 improvement, whereas increasing the queries almost leads to an approximate 1 point of F1 increase.

**Depth order recovery.** We report the results when varying the backbone of the segmentation backbone from Swin-T$_{100}$ to Swin-L$_{200}^{\ddagger}$ in Tab. 4b. These results show very different trends with respect to the occlusion order backbone ablation. For Mask2Order$^o$ networks, they seem to hastily plateau around 18.75 WHDR (all) with slight improvement from Swin-T$_{100}$ to Swin-B$_{100}^{\ddagger}$. Increasing the number of queries from 100 to 200 slightly fosters performance. We note the positive effect of simultaneously performing both the task of occlusion and depth predictions as the results of Mask2Order$^{o,d}$ Swin-T$_{100}$ already overcomes those of Mask2Order$^o$ Swin-L$_{200}^{\ddagger}$. Moreover, Mask2Order$^{o,d}$ Swin-L$_{200}^{\ddagger}$ beats its Mask2Order$^o$ counterpart by 1.43 points of WHDR (all).

**Task difficulty.** We observe that the *holistic* task of depth order prediction seems to be easier to perform than the task of occlusion order prediction since the WHDR discrepancy from InstaOrderNets and InstaFormer is larger than their F1 difference (cf. Tab. 1 of the main paper) and since changing from a weaker to a stronger backbone mostly benefits the occlusion order prediction networks (from Tab. 4a and Tab. 4b).

A key property of depth matrices is that they always contain $\frac{n \cdot (n-1)}{2}$ positive values since objects must be one behind the others. This is not the case in occlusion order prediction, where no *prior* on instance layout exists. Moreover, depth is consistent across instances. Consider $A$ and $B$ two instances of the image. If we have $A \to B$, then, $A$ is in front of $B$ and thus we know that we can transfer all the depth relations of $A$ to $B$. On the other hand, this property does not apply to occlusion order prediction.

Note that this property only emerges when switching to the *holistic* paradigm since we can draw information from all instances at once. We believe this property is important in explaining the strong results for depth order prediction and saturation across backbone sizes.

### E.2 Adapters

We ablate the adapters of InstaFormer models. The results of these experiments are reported in Tab. 5.

**Occlusion order recovery.** The main observation is that adapters are important for obtaining strong performances on the occlusion order prediction benchmark. Most notably, InstaFormer[o] benefits from an F1 improvement of 2.18. We observe that most of that result is explained by the increase in recall.

**Depth order recovery.** The depth ordering benefits the most from the use of adapters. As a matter of fact, InstaFormer[o,d] shows a spectacular 10.10 improvement of WHDR all. This result is explained by a sharp reduction in both distinct and overlap WHDR. We observe similar trends for InstaFormer[d].

## F  Inference cost

### F.1  Runtime comparison

We benchmark the runtime inference of our *holistic* InstaFormer and compare it against *pairwise* InstaOrderNet, in Fig. 5a of the main paper. Runtime for our proposed InstaFormer appears constant with respect to the number of instances in the image, while InstaOrderNet is exponential. Moreover, InstaFormer's efficiency is amortized for any image containing more than 7 instances and up to 12 times faster than the baseline when the image contains 20 instances. This makes our approach suitable for prediction on natural images since such scenes usually contain many objects.

We also provide an analysis on a batched version of InstaOrderNet where the pairwise relations are sent to the network in a single batch. While this approach is the fastest, we highlight its high memory requirements, which makes it impractical for users with limited resources or embedded systems.

### F.2  Memory consumption

We further benchmark the memory consumption of our InstaFormer[o,d] network and compare it to InstaFormer[o,d] in Fig. 5b of the main paper. The memory consumption of our InstaFormer grows linearly with the number of instances on the image, while InstaOrderNet's remains constant. This highlights the trade-off between inference speed and memory consumption. We argue that InstaFormer still wins that trade-off since its runtime is constant while InstaOrderNet's is exponential.

We also provide an analysis on a batched version of InstaOrderNet, where the pairwise relations are sent to the network in a single batch. We observe that the memory requirements quickly become prohibitive. In fact, we encounter a CUDA out-of-memory error for any image containing more than 20 instances. We thus stop the benchmarking at this threshold. This is a critical drawback, as natural scenes usually depict a large number of instances at once.

### F.3  Scalability

In Tab. 6, we compute the number of average instances detected in images as well as the number of maximum predicted instances in a single sample across the validation set of COCO [26]. We vary the number of object queries, but we observe in both cases that the number of instances predicted in a sample is close or above to 11. Comparing this result to Fig. 5a and Fig. 5b, we conclude that it is *on average* better to use InstaFormer than other baselines since natural images contain, *on average*, close to, or above 11 instances.

The second column of Tab. 6 emphasizes the fact that our InstaFormer network can scale way above 10 instances although the training set of INSTAORDER is limited to annotations containing 10 objects [24]. As a matter of fact, InstaFormer with 200 object queries detects 63 instances on an image of the COCO validation set. In both 100 and 200 object queries settings, the value of maximum instances detected is far below the total number of object queries. This leads us to conclude that InstaFormer is practical for general inferences in natural scenes.

| Num. Queries | Avg. Predicted Instances (per sample) | Max. Predicted Instances (in a single sample) |
|---|---|---|
| 100 | 10.91 | 56 |
| 200 | 11.91 | 63 |

Table 6: **Inference statistics of InstaFormer**. We compute the number of average predicted instance per samples and the number of max predicted instances in a single sample on the full COCO validation for model with different number of object queries.

# G   Qualitative results

## G.1   Comments on results from the main paper

In this section, we comment on qualitative results from the main paper in more detail.

In Fig. 1 of the main paper, we display two in-the-wild results. In the "athletics" example, we note the occlusion from the hand of "Person1" to "Person3" that was effectively captured from our model. Note that this occlusion is difficult to notice without carefully observing the input RGB image.

Still in that figure, the "table tennis" example shows interesting bidirectional occlusion examples where the handling of the rackets are properly related to their players in the occlusion graph. Note that in both cases, the depth graph, while crowded on the second example, also depicts accurately the layout of the instances in the scene. This is obvious for the "athletics" example. For the "table tennis" example, careful inspection reveals no sign of unrealistic depth ordering.

Analyzing Fig. 4, in the main paper, the first row contains "elephants" and "K-Pop", which are both in-the-wild predictions from our network. We observe an ambiguous occlusion case in the "elephants" example, where "Elephant3" does not seem to occlude "Elephant2" even though reported as such on the occlusion graph. While it is ambiguous whether "Elephant3" is closer to the camera than "Elephant1", the network still appears to predict a coherent order.

The second image of the first row of Fig. 4, namely "K-Pop" is an impressive example of accurate depth order prediction in a complex scene. While the input RGB image solely depicts silhouettes, our InstaFormer network still predicts an accurate and coherent depth layout for the scene. In that case, the network shows difficulties understanding the occlusion orderings for some elements since there are little clues to rely on. For example, "Person6" and "Person4" are linked with a bidirectional occlusion even though it seems that it should be unidirectional from "Person4" to "Person6". The same holds for "Person1" and "Person3".

All the remaining images of Fig. 4 are extracted from the validation set of INSTAORDER [24]. We specifically highlight the "donuts" example to emphasize that the model can still predict accurate orderings even crowded scenes.

On the second row and second column, the "plate" example highlights a perfect prediction in a simple scenario.

On the last row of Fig. 4, the "cooking cat" example shows that our model can predict sane relations even if the "Cat" is blurred. As a matter of fact, the cat indeed occludes the "Oven", yet the "Oven" does not. Careful inspection reveals that the blurriness of the cat lets appear a notch of the "Oven" that *visually* occludes the "Cat" even though it is physically impossible.

Finally, the last example on the last row and last column of Fig. 4, called "stop sign", shows a neat case of a prediction of perfect occlusion and depth orders, even though the size of the cars is noticeably small. We also emphasize this result as no occlusion between elements occurs. We recall that in COCO [26], the stop sign class is only annotated on the sign itself, and not the pole.

## G.2   Ground-truth visualizations

We visualize the ground-truths for the INSTAORDER validation set prediction of Fig. 4 in Fig. 9 and Fig. 10 on Fig. 11. Note that, in some rare cases, our network was unable to detect some of the instances in the image, leading to no ordering prediction (Fig. 9, "donuts", "cat in the oven"). However, more frequently, it predicted more instances than the ground-truth annotation, for example, "sheep2" is accurately detected along with its proper occlusion ordering (first row, last column of

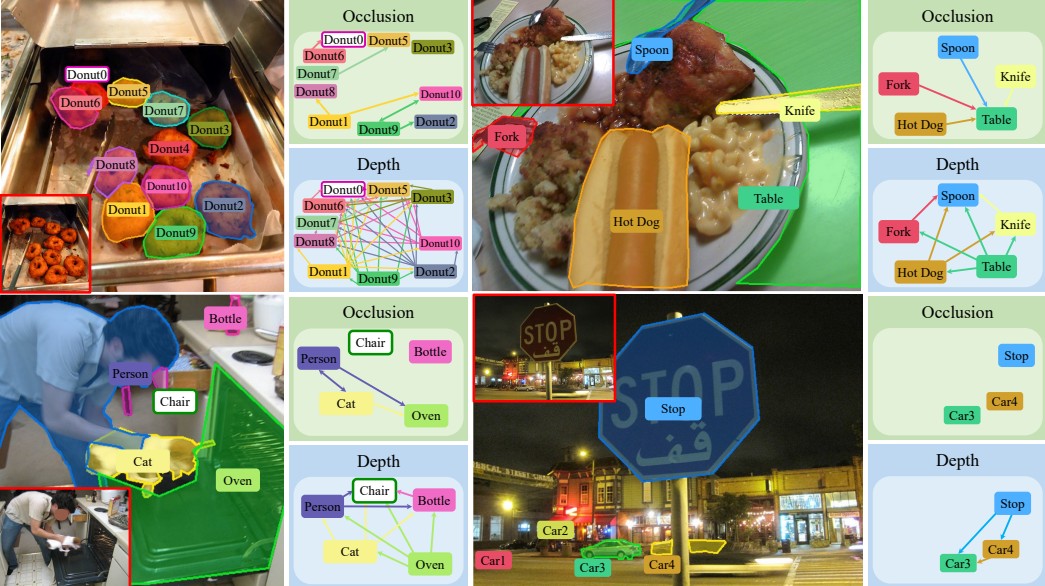

Figure 9: **Ground-truth visualization of the predictions from Fig. 4**. We visualize the order graphs and the masks from the INSTAORDER dataset and overlap their respective COCO labels on top of them. However, we re-use the same colors as the predicted classes from InstaFormer in Fig. 4 for better comparison with the predictions. We use a white box with colored border to indicate ground-truth that have not been matched from our network's predictions. Annotated objects on the image not present in the graphs symbolizes that our network predicted these nodes even though not present in the ground truth.

Fig. 10 and Fig. 11). This is also the case for the teddy bears (see Fig. 10 and Fig. 11). Sometimes, our network predicts even better orderings than the ground-truth, for example, in Fig. 10, in the cake picture (third row, second column), the network predicts a bidirectional occlusion between "Person3" and "Cake". This seems more reasonable than the ground-truth annotating a unidirectional order (Fig. 11). In some cases, the objects are too thin to be detected, in which case the predicted orderings are still sane, although they could probably be more precise ("kite", 4th row, 2nd column on Fig. 10 and Fig. 11)

### G.3 More qualitative results

We provide even more qualitative results obtained from InstaFormer[o,d] on INSTAORDER in Fig. 10. And report their ground-truths in Fig. 11.

## H Web image sources

We provide the link to all the web images used in this paper.

Figure 1 in the main paper contains two images, both of which were extracted from the web. Left image "table tennis". Right image "athletics".

Figure 4 of the main paper is composed of 3 rows, out of which only the first one is composed of web images. First row, left image "elephants". First row, right image "K-Pop".

All the remaining images present in our paper originate from the COCO-stemmed INSTAORDER dataset [24, 26] (specifically from the validation set), which is licensed under a Creative Commons Attribution 4.0 License.

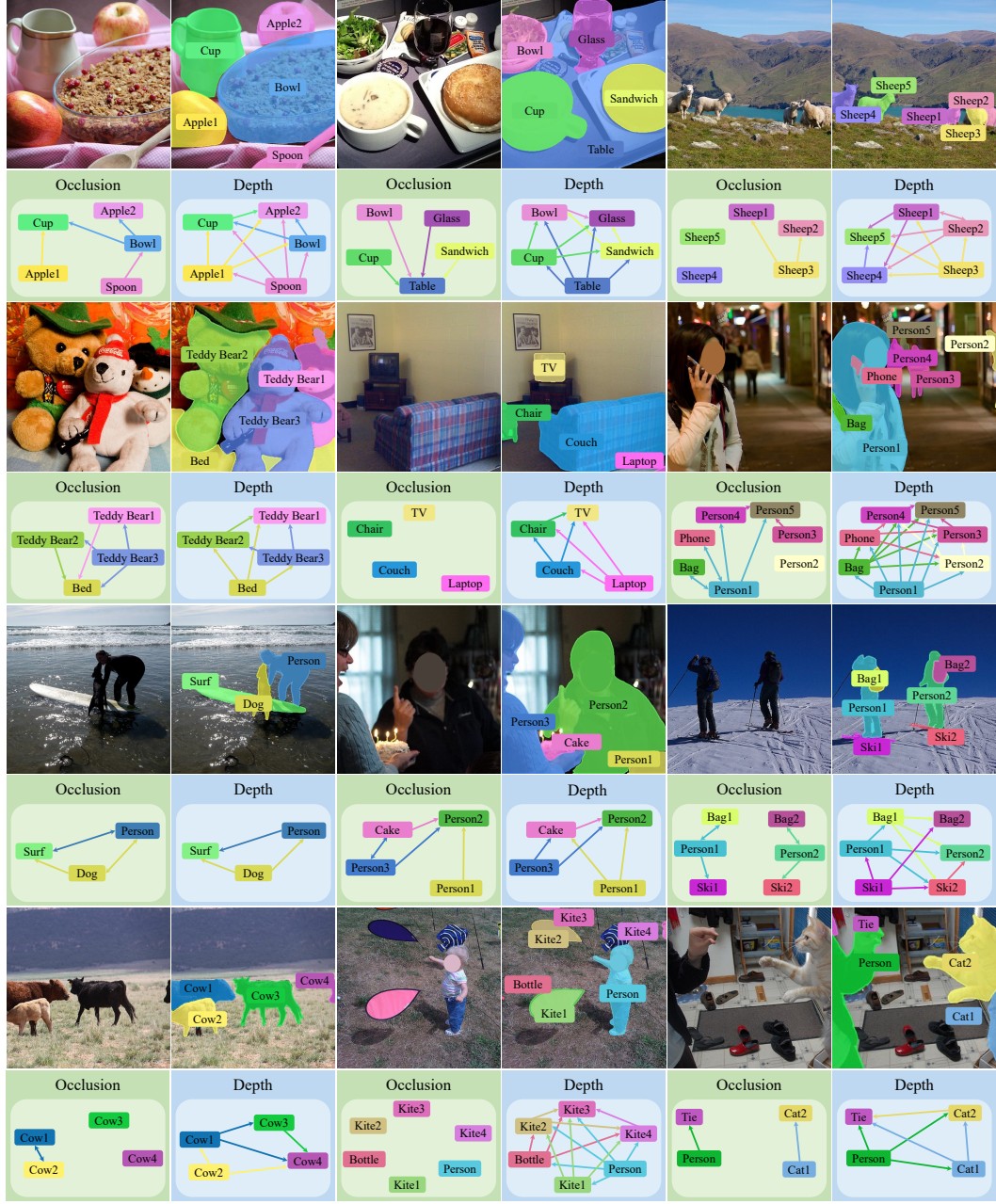

Figure 10: **More qualitative results for InstaFormer[o,d]**. All the images are from the validation set of INSTAORDER.

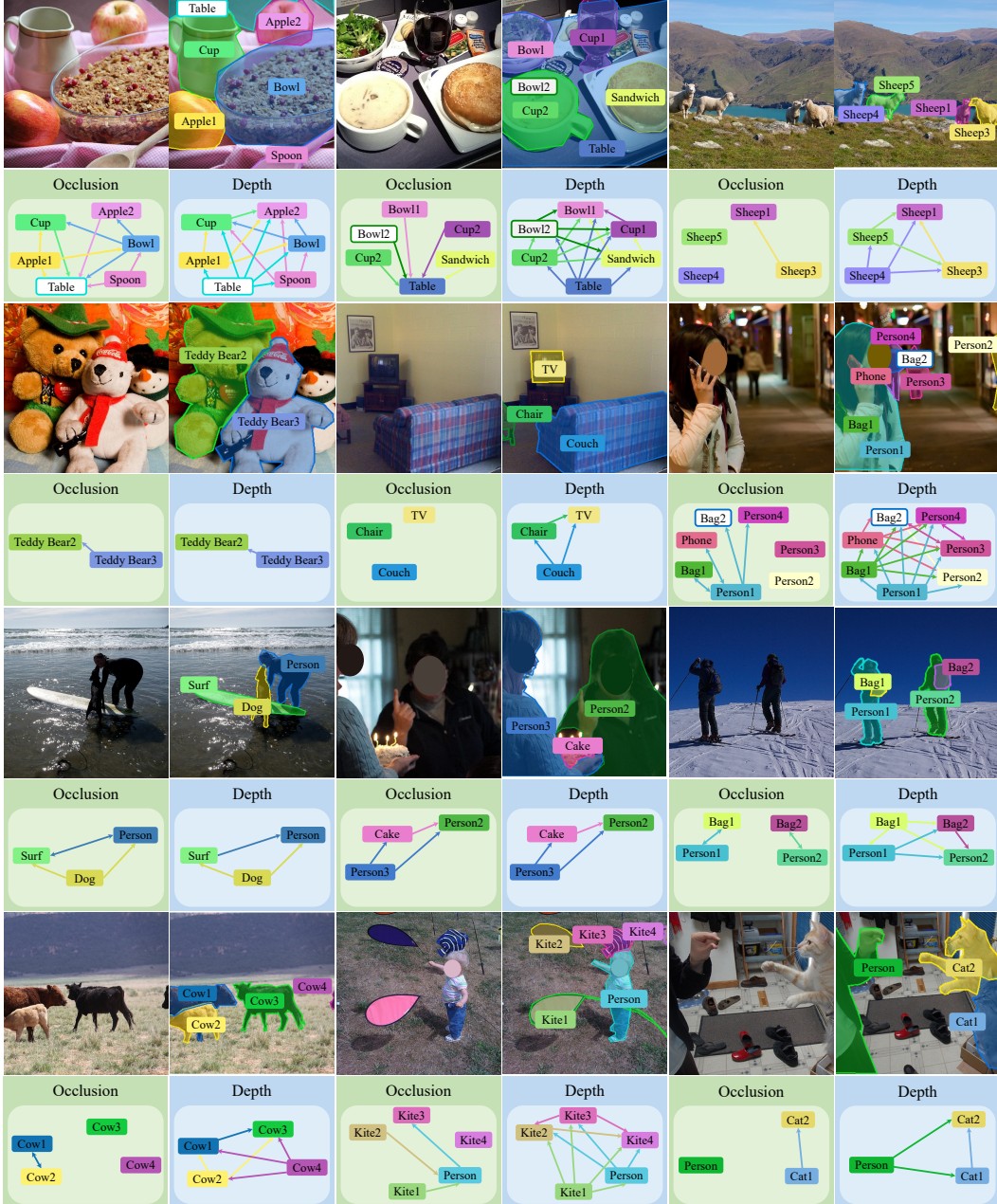

Figure 11: **Ground-truth visualization of the predictions from Fig. 10**. We visualize the order graphs and the masks from the INSTAORDER dataset and overlap their respective COCO labels on top of them. However, we re-use the same colors as the predicted classes from InstaFormer in Fig. 10 for better comparison with the predictions. We use a white box with colored border to indicate ground-truth that have not been matched from our network's predictions.

