# OpenReview forum: "Holistic Order Prediction in Natural Scenes"
_NeurIPS.cc/2025/Conference — NeurIPS 2025 poster_

### Official Review · Reviewer_i3cT · 2025-06-23

**Clarity:** 1
**Significance:** 3
**Originality:** 2
**Rating:** 4
**Confidence:** 4

**Summary:**

The paper proposes a novel approach for occlusion and depth ordering of objects in a scene. In particular, while most approaches do this by evaluating pairs of objects in each forward pass, the authors propose a method for "holistic" ordering, meaning that a single forward pass is able to construct the whole ordering matrix. The approach is validated on a common dataset, and the authors test it against several baselines, from specialised to naive ones, and perform a number of ablation studies to prove the effectiveness of the approach.

**Questions:**

Major:
* More details on the task in Sec. 3.1. For example, on how the matrix G is usually constructed (differences between g_ij and g_ji? How does g_ij look like if object i and object j are completely unrelated? etc.)
* More training details (loss functions unspecified, mentioned some auxiliary ones in line 252 without specifying which one and for what purpose, etc.), as the paper should allow people to reproduce the approach in its entirety, and it's currently not the case.
* Quantitative analysis on efficiency of "holisticity": report inference time and/or GFLOPs of proposed approach and baselines.

Minor:
* Some figures/tables are not hooked in the text (Fig.1, Fig. 2b, etc.)
* Contributions in the introduction unclear (Contrib #2 is unclear and should be reformulated, Contrib #3 is arguably not a contribution at all)
* Include appendix at the end of the paper?
* Why are the baselines in Sec. 3.3 discussed in the main section dedicated to the proposed approach?

**Ethical Concerns:**

["NO or VERY MINOR ethics concerns only"]

**Final Justification:**

First, I want to thank the authors for a thorough response to all reviewers, including me.
The authors did a good job in addressing all reviews and convincingly motivated their choices. Additionally, they promise to better specify all parts I marked as unclear.
My opinion on the paper (even after a thorough re-read) is positive, as it tackles an interesting topic and proposes a novel approach that can be relevant for the community, and was mainly hindered, as I specified, by the lack of clarity.
Given the rebuttal phase and the authors' responses, I'm willing to increase my score from 3 to 4. I strongly recommend the authors to integrate all changes from all discussion in the final version of the paper.

**Limitations:**

Limitations are not really addressed, as something is mentioned in the introduction but it's mainly a limitation of the existing dataset to evaluate the approach. A proper evaluation of the limitations of the proposed approach is necessary. A (rather short) sentence on possible future works in, instead, present.

**Paper Formatting Concerns:**

No formatting concern.

**Quality:**

2

**Strengths And Weaknesses:**

The paper tackles an interesting topic, relevant and relatively under-explored so far. The solutions the authors come up with are interesting and mostly well-motivated, and the experimental evaluation is convincing.

The main limitation of the paper is that it is unclear. Many parts are hard to understand, and overall the approach is not clear once a reader goes over Sec. 3. Additionally, more details on the task itself could be shared, as it is a relatively uncommon task (for now), and not as easily-recognizable as other CV tasks. Also, the authors repeat several times that their holistic approach is more efficient as it predicts everything in a single forward pass, but its efficiency compared to the other baselines is not tested quantitatively.

---

> ### Author Rebuttal · Authors · 2025-07-31
>
> # Authors' response to reviewer i3cT
>
> We deeply thank reviewer i3cT for their detailed and constructive comments. We are grateful towards
> the reviewer for recognizing the comprehensiveness of our benchmark and the efficiency of our
> approach. We appreciate the reviewer mentioning that while this task is under-addressed and
> constitutes an interesting topic of research. We address the reviewer's remarks as follows:
>
> ## Weaknesses
>
> > Many parts are hard to understand.
>
> We thank reviewer i3cT for their honest feedback. We would be happy if they could point out precise
> elements that need clarification (phrasing of sentences, the terminology, the notation, etc.) so
> that we can incorporate it into the final version of the paper.
>
> > Approach is not clear after reading section 3.
>
> Similarly, we are willing to rephrase, re-write, re-explain or add more details in
> section 3 upon specific requests. Before we release the final version, we will make sure to read
> and re-write this section to make it as clear as possible.
>
> > More details on the task could be shared as it is relatively uncommon in the CV community.
>
> Although our baseline papers:
>
> - Self-supervised scene de-occlusion. Xiaohang Zhan et al.
> - Amodal instance segmentation with kins dataset. Lu Qi et al.
> - Instance-wise occlusion and depth orders in natural scenes. Hyunmin Lee et al.
>
> are all published in CV conferences, we agree that the task has not been yet deeply explored.
> **We are willing to polish our paper and will include more details on the task and the dataset
> format in the supplementary materials** of our final version (cf. "questions" section for more
> details).
>
> > The authors mention that their holistic approach is more efficient as it predicts everything in
> > a single forward pass, but its efficiency compared to other baselines is not tested quantitatively.
>
> **We report the inference time and the memory consumption of our proposed proposed InstaFormer
> and compare it against baselines in section F and figure 7 of the supplementary materials**.
> We choose InstaOrderNet as the baseline for comparison since it constitutes the most recent
> and competitive approach while being representative of all pairwise approaches. We benchmark the
> official implementation of InstaOrderNet, but also implement a batched version of it that
> consists in stacking all the pairs of instances in a single batch and performing inference on them
> all at once. We refer to this as InstaOrderNet (batched). Note that InstaFormer is not
> optimized for speed.
>
> In figure 7a, **InstaFormer's inference time appears constant**, while
> InstaOrderNet's grows exponentially with respect to the number of predicted instances in the
> scene. InstaFormer's inference time averages below 1 second and becomes advantageous over
> InstaOrderNet's default inference anywhere above 11 instances. The batched version of
> InstaOrderNet consistently outperforms InstaOrderNet and InstaFormer.
>
> **Yet, in practice, it is hard to use the batched version of InstaOrderNet since it requires an
> exponential amount of memory with respect to the number of predicted instances.** As a matter of
> fact, while InstaOrder's default implementation runs in a constant memory size, its batched
> inference quickly becomes prohibitive. Inference for more than 20 instances results in more than
> 40Gb of memory usage. On the other hand, **InstaFormer's memory usage grows linearly** with respect
> to the number of instances. It proves more affordable than InstaOrderNet's batched inference for
> more than 10 instances and **saves almost 30Gb of memory for 20 instances**. We encounter CUDA OOM
> error for any inference containing more than 20 instances using InstaOrderNet (batched), point
> at which we stop the experiment.
>
> **We believe that these results place InstaFormer in a favorable position considering the runtime
> speed and memory trade-off**. Further optimization of the inference procedure can decrease
> its runtime even further.
>
> ## Questions
>
> ### Major
>
> > How is $G$ usually constructed?
>
> For both occlusion and depth order prediction $G$ represents the adjacency matrix of the order
> relations between the instances in the scene. The rows and columns each represent an instance
> while the entries represent the order relations (occlusion or depth) between them.
>
> Given $I \in \[2, \dots, N\]$ instances in the scene, we consider a pair of instances
> $(i, j) \in I \times I$ such that $i \neq j$ and consider their entry in the order matrix
> $G_{i,j}$. We never consider the case where $i = j$, i.e. the diagonal of the matrix, since it
> does not make sense for an instance to occlude (resp. be in front/behind) itself. Thus, we always
> manually fill the matrix diagonal with -1 (this value is arbitrary).
>
> The occlusion order task predicts binary labels $G_{i,j} \in \lbrace0,1\rbrace$ for all $(i, j)
> \in I \times I$, such that $i \neq j$, symbolizing whether:
>
> 1. instance $i$ does not occludes instance $j$ (label 0).
> 2. instance $i$ occludes instance $j$ (label 1).
>
> In the case where $G_{i,j} = G_{j,i} = 1$, we say that there is a bidirectional occlusion relation
> between $i$ and $j$.
>
> The depth order task predicts a ternary label $G_{i,j} \in \lbrace0,1,2\rbrace$ for all $(i, j)
> \in I \times I$, such that $i \neq j$, symbolizing whether:
>
> 1. instance $i$ is behind instance $j$ (label 0).
> 2. instance $i$ is in front of instance $j$ (label 1).
> 3. instance $i$ and $j$ share the same depth region (label 2). This is the case when the two
>    objects span in a large overlapping portion of the 3D space.
>
> **We will complement the paper with this explanation in the final version.**
>
> > What is the difference between $G_{i,j}$ and $G_{j,i}$?
>
> $G_{i,j}$ represents the order relation from $i$ to $j$ while $G_{j,i}$ represents the order
> relation from $j$ to $i$.
>
> For occlusion, this is because the occlusion relation is not always symmetric:
> bidirectional occlusion relations are possible, but not always present. Sometimes, only $i$ might
> occlude $j$ and sometimes, only $j$ might occlude $i$.
>
> Depth on the other hand is symmetric. An instance if $i$ appears in front of $j$,
> then $j$ must necessarily appear behind $i$, unless they share the same depth range, in which case
> they are both annotated using label 2.
>
> > How does $G_{i,j}$ and $G_{j,i}$ look like if object $i$ and $j$ are completely unrelated?
>
> Naturally speaking, the notion of "unrelated" only applies to the occlusion order task. In the
> depth order task, all instances are "related" since they all necessarily appear one behind the
> others.
>
> For the occlusion order task, we understand the notion of "unrelated" as a case where $i$ does not
> occlude $j$ and vice-versa, thus, their entries in the order matrix would simply be $G_{i,j} =
> G_{j,i} = 0$.
>
> > More training details (loss, etc.).
>
> We have overlooked the details of our training recipe and we thank reviewer i3cT for offering us
> the opportunity to clarify it.
>
> - Losses
>   - Occlusion order optimization: we use the **binary cross-entropy (BCE)** loss.
>   - Depth order optimization: we use the **cross-entropy (CE)** loss.
>   - Occlusion + depth order optimization: we use both BCE and CE losses, equally weighted.
>
> We add auxiliary losses to help convergence of our networks. We detail them below:
>
> - Auxiliary losses
>   - Occlusion order: we add a BCE loss on the occlusion order logits of all transformer
>     decoder layers.
>   - Depth order: we add a CE loss on all the transformer decoder layers.
>   - Occlusion + depth: we add both BCE and CE losses on all the transformer decoder layers.
>
> - Batch size
>   - 16 in all cases.
>
> We hope that these clarifications are sufficient to help the reviewer understand our training setup
> and full reproduction of our experiments. **We will include all those details in the revised version
> of our paper**. We will also **provide a link to our repository upon acceptance, allowing to
> re-train all InstaFormer networks in a single command.**
>
> > Quantitative analysis on efficiency of "holisticity", i.e. inference time and memory of proposed
> > approach and baselines.
>
> We kindly refer reviewer i3cT to our answer in the "weaknesses" section above and to the
> supplementary section F along with figure 7.
>
> ### Minor
>
> > Some figures are not hooked in the text.
>
> We thank reviewer i3cT for pointing out this issue. We will hook all figures to the text in the
> final version of the paper.
>
> > Contributions in the introduction are unclear (\#2 is unclear, \#3 is arguably not a
> > contribution).
>
> **We will drop both contribution \#2 and \#3 in the final version of the paper**. We will
> **polish the first item of the contributions** to reflect the nature of our submission.
>
> > Include appendix at the end of the paper?
>
> We agree with reviewer i3cT. **We will move the appendix to the end of the paper in the final
> version.** We do not see any reason not to merge it with the main paper now that it has been
> finalized.
>
> > Why are the baselines in Sec. 3.3 discussed in the main section dedicated to the proposed
> > approach?
>
> **We agree with the reviewer after reading the paper again. We propose to move the baseline
> discussion to the experiment section (sec. 4) of the paper to keep the flow focused on our
> InstaFormer approach in Sec 3.** We welcome any further suggestions on how to improve the flow
> of the paper.
>
> ## Limitations
>
> > Limitations are not really addressed. A proper evaluation is necessary.
>
> **We highlight a failure case of our InstaFormer network in section G.3 and figure 9 of the
> supplementary materials**. We call this failure case "instance mix-up". We kindly refer the
> reviewer to it. **We also observed another failure case which we call "segmentation failure"**
> where mispredicted masks lead to creating nodes in the graph that belong twice to the same
> instance, resulting in partially wrong order predictions. **We will include this failure case in
> the final version of the paper.**

---

> > ### Author Response · Authors · 2025-08-06
> >
> > Dear reviewer i3cT,
> >
> > We hope this message finds you well. We are writing to kindly follow up on our manuscript rebuttal.
> > We would be grateful for any feedback at your convenience. We hope that our rebuttal have
> > addressed questions and concerns you may have had. We would be more than happy to provide any
> > additional information or clarifications if needed.
> >
> > Thank you very much for your time and consideration.
> >
> > Best regards,
> >
> > > Authors of Submission9867

---

> > ### Author Response · Authors · 2025-08-09
> >
> > # Authors' comprehensive summary
> >
> > Dear reviewer i3cT, we hope this message finds you well. We are thankful for your detailed
> > review and for your comments on our manuscript. We believe that this helped us refine it
> > significantly and results in a comprehensive and intuitive presentation of our work.
> >
> > > Many parts are hard to understand and approach is not clear after reading section 3.
> >
> > We continuously refined the manuscript to improve clarity and readability
> >
> > > More details on the task.
> >
> > We also revised the task presentation, which now should become easier to follow for reviewers
> > who are exposed to the task for the first time.
> >
> > We also provided many examples of use cases for the depth order task in our answer to
> > reviewer SiaZ, which we believe could also help you understand the task better.
> >
> > This change will be reflected both in the main paper of the manuscript and also in the
> > supplementary material.
> >
> > > Construction of $G$ and relation between $g_{ij}$ and $g_{ji}$.
> >
> > We provided a detailed description of the graph $G$ and the relation between $g_{ij}$
> > and $g_{ji}$ in our rebuttal.
> >
> > > Training details.
> >
> > We added more details on the training procedure, including the batch size, objectives, etc.
> > in our reply.
> >
> > > Quantitative analysis of "holisticity".
> >
> > We led an experiment in the supplementary material (sec. F, fig. 7) to observe the runtime
> > performance and memory consumption of our method compared to the baselines. We conclude
> > that we achieve the best trade-off between performance and efficiency.
> >
> > Reviewer NnW7 also raised a similar question, that we addressed in detail. In the event
> > to learn more, we would kindly refer you to this rebuttal message.
> >
> > > Minor considerations
> >
> > We fixed all the minor considerations that you raised in the review, including moving
> > the baselines discussion to the experiment section.
> >
> > We hope that in the light of these explanations and changes, you will find our approach
> > convincing and manuscript satisfactory. We are always open to further discussions and
> > clarifications before the deadline of the discussion phase.
> >
> > We would once again express our gratitude towards the time you spend on reviewing NeurIPS
> > submissions.
> >
> > Best regards,
> >
> > > Authors of Submission9867

---

> ### Comment · Area_Chair_U6xq · 2025-08-07
> **Please read the rebuttal and discuss with the authors**
>
> Dear Reviewer i3cT,
>
> Thanks for serving for NeurIPS. Please read the rebuttal and share your thoughts here. The author-reviewer discussion deadline is tomorrow. Please note that per NeurIPS's policy, you have to post a discussion before clicking on the mandatory acknowledgement.
>
> Thanks,
>
> AC

---

### Official Review · Reviewer_NnW7 · 2025-07-01

**Clarity:** 2
**Significance:** 2
**Originality:** 2
**Rating:** 4
**Confidence:** 3

**Summary:**

This paper proposes a solution for simultaneously predicting occlusion and depth orders of all instances in a scene from a single RGB image in a single forward pass. The key novelty lies in predicting the ordering matrix ​**G**​ in a single forward pass, thereby eliminating the need for pairwise inference.

**Questions:**

I must acknowledge that I am not an expert in this field; however, after reading this paper, I have the following questions:

- **Scalability:** Can your method handle more than 10 instances? Handling fewer than 10 instances appears impractical. The maximum number of instances might be constrained by the value of N. Perhaps incorporating more tokens could address this.

- **​Inference Efficiency:** How does the speed of your solution compare to pairwise methods? For instance, reporting Frames Per Second (FPS) on 1024×1024 images would be beneficial. Pairwise methods might achieve faster inference through batch processing, potentially outperforming the original approach. Therefore, it's essential to ensure that comparisons are fair and that both implementations are optimized appropriately.

- **​Readability of Figure 3:** The rationale behind designing the method as depicted in Figure 3 is not entirely clear. For example, could you elaborate on the rationale behind using a transformer with only one layer? There may be underlying motivations that make your design unique and theoretically superior to alternatives. It would be helpful to elaborate on these motivations before delving into the specifics of the design.

- **Theoretical Advantages of Predicting ​G​ in One Pass:** Does predicting the ordering matrix ​G​ in a single forward pass offer any theoretical guarantees of superiority over previous methods in terms of accuracy, independent of empirical results? It would be beneficial to understand if there are inherent theoretical benefits that position this approach as fundamentally more accurate or robust compared to traditional pairwise inference techniques.

**Ethical Concerns:**

["NO or VERY MINOR ethics concerns only"]

**Final Justification:**

The authors have effectively addressed all of my concerns.

**Limitations:**

The paper does mention certain limitations of their approach. However, it would be beneficial if the method could handle an unlimited number of instances, thereby increasing its practicality and effectiveness in more complex scenarios.

**Paper Formatting Concerns:**

I reviewed the paper and noticed no major formatting issues.

**Quality:**

2

**Strengths And Weaknesses:**

**Strengths:**
The paper show state-of-the-art (SOTA) performance relative to pairwise methods.
It presents seemingly comprehensive experimental results and ablation studies.
​

**Weaknesses:**
The computational efficiency claims lack empirical validation, such as frame per second (FPS) metrics or memory usage, which are essential for assessing practical applicability. The scalability of the method beyond the dataset's 10-instance limit remains unverified, raising questions about its real-world applicability in scenarios with a larger number of instances.

---

> ### Author Rebuttal · Authors · 2025-07-31
>
> # Authors' response to reviewer NnW7
>
> We are pleased to hear that reviewer NnW7 appreciates our work and points out the strong and
> comprehensive results of our approach while adopting a novel design. We thank reviewer NnW7
> for its precious comments and we address them as follows:
>
> ## Weaknesses
>
> > The computational efficiency claims lack empirical validation such as FPS or memory usage,
> > essential for assessing practical applicability.
>
> **We ran a computational efficiency comparison experiment in section F of the supplementary materials**.
> We kindly refer reviewer NnW7 to the aforementioned section and figure 7 of the supplementary
> materials for a detailed analysis of the computational efficiency of our approach compared to
> the InstaOrderNet baseline.
>
> Nonetheless, we will quickly summarize the results. We compared InstaFormer against the
> InstaOrderNet baseline since this is the most recent art and competitive approach. We used the
> official implementation of InstaOrderNet but also realized that there is a more efficient
> inference procedure that consists in batchifying all the pairs of instances together and
> performing inference on them in a single batch. We refer to this as InstaOrderNet (batched).
> For all InstaOrderNet variants, we resize the image to 256x256 pixels. For InstaFormer, we
> follow Mask2Former's inference script and do not resize the images. Please, note that this
> results in a disadvantage for InstaFormer since the images have a larger resolution.
>
> We compute inference speed in seconds and memory usage in Gb with respect to the number of
> instances in the order graph.
>
> **For inference speed, we observe that InstaFormer appears with a constant inference time**, while
> InstaOrderNet appears exponential with respect to the number of instances. Our approach's average
> inference time lies below 1 second. We emphasize the fact that we have not yet optimized the method
> for speed. Above 11 instances, InstaFormer becomes faster than the default InstaOrderNet. On the
> other hand, the batched InstaOrderNet is consistently faster than both InstaOrderNet and
> InstaFormer.
>
> However, this batched performance comes at a cost. Comparing the memory usage, we observe that inference
> on the batched InstaOrderNet rapidly becomes prohibitive. As a matter of fact, its memory
> complexity grows exponentially with respect to the number of instances in the scene. **On the
> other hand, InstaFormer shows a linear memory growth. Above 10 instances, InstaFormer's cost
> falls below this of InstaOrder (batched). At 20 instances it saves almost 30Gb of memory compared
> to the batched version of InstaOrderNet**. We encounter CUDA OOM errors for any inference above
> 20 instances using InstaOrderNet (batched), thus we stop the benchmark here. Naturally, the
> default InstaOrderNet memory usage remains constant with respect to the number of instances.
>
> We believe that these results show that **InstaFormer obtains the best trade-off between runtime
> speed and memory usage**, especially when instances in the scene grow beyond 10. Further
> optimizations of the inference method could lead to even better results.
>
> > The scalability of the approach beyond the dataset's 10 instances limit remains unverified.
>
> When benchmarking our approach in figure 7 of the supplementary materials, we benchmark results
> up to 20 instances in the scene which supports the scalability of our approach.
>
> We also empirically observe that **our approach scales well largely above 10 instances by
> predicting the order graphs on the whole validation set of COCO**. We report the average and
> maximum number of instances returned by InstaFormer^o,d in the following table:
>
> |Num. Queries|Avg. Predicted Instances (per sample)|Max. Predicted Instances (in a single sample)|
> |:-|:-:|:-:|
> |100|10.91|56|
> |200 |11.91|63|
>
> In all cases, **we observe that we are not bounded by the amount of queries we use to predict the
> order graphs**. Moreover, we showed the computational advantages of using InstaFormer over
> baselines when the number of instances in the scene grows beyond 10 in the previous question.
> This is on average the case in COCO's natural scenes.
>
> ## Questions
>
> We thank reviewer NnW7 for their important questions. We address them as follows:
>
> > Scalability.
>
> > Inference efficiency.
>
> Please, refer to the previous section in which we address both these items.
>
> > Readability of Fig. 3. Could you elaborate on the rationale behind using a transformer with only
> > one layer? There may be underlying motivations that make your design unique and theoretically
> > superior to alternatives. It would be helpful to elaborate on these motivations before delving
> > into the specifics of the design.
>
> We thank reviewer NnW7 for offering us the opportunity to clarify the design choices of our
> proposed InstaFormer network.
>
> ### Task formulation
>
> The main motivation behind our method is to **reformulate the problem of pairwise order prediction
> as a full adjacency matrix prediction**. We can now think about the problem in reverse order:
> if we start from the adjacency matrix of the order graph $G$, we can think of each element $G_{ij}$
> as the result of an **order compatibility score between the instances $i$ and $j$**. We can then
> model this score by performing a simple dot product over tokenized representations of $i$ and $j$.
>
> Since the transformer inherently behaves at the token level, the choice of using a transformer
> naturally emerges at this point. Yet, there remains a question: "how to encode the instances into
> tokens?".
>
> ### Instance encoding
>
> Ablation in table 2a of the main paper shows that simply using the queries of Mask2Former as
> input is not an optimal choice. However, we Mask2Former's per-pixel decoder latent features are
> aligned with the query instances of the transformer (i.e. represent the same instances) while
> containing complementary information. These features are crucial determining delicate order
> predictions since they are pixel-aligned. Thus, we create descriptors from the per-pixel latent
> space of Mask2Former and observe (still in table 2a of the main paper) that this formulation of
> the compatibility score results in the best performance.
>
> We would like to mitigate any ambiguity by recalling that the main transformer decoder of InstaFormer
> contains 8 layers. However, reviewer NnW7 is right in pointing out that the encoder for the
> instance-aligned per-pixel features of our geometrical order head contains a single transformer
> layer.
>
> We empirically observed that simply max-pooling the features results in lower performances (cf.
> table 2b of the main paper). **Adding a transformer layer dramatically increases the performance
> of the model**. Our motivation for adding this layer resides in the fact that **the
> instance-aligned per-pixel features of Mask2Former have not been able to share critical spatial
> information used by InstaFormer to determine the order relations between instances.**
> Self-attention at the instance level allows the model to generate much more comprehensive and
> robust instance descriptors that, in turn, lead to better performance across occlusion and depth
> order accuracy.
>
> Thanks to reviewer NnW7's comment, we extended this ablation and increased the number of
> transformer layers before the max-pooling operation. We present the results in the following table:
>
> |Pooling strategy|Precision|Recall|F1|WHDR (distinct)|WHDR (overlap)|WHDR (all)|
> |:-|:-:|:-:|:-:|:-:|:-:|:-:|
> |Max-Pool|74.86|87.87|79.01|16.08|37.87|21.88|
> |Transformer-1L+Max-Pool|**88.64**|75.56|79.74|**8.43**|**25.36**|**14.03**|
> |Transformer-2L+Max-Pool|87.62|**78.30**|79.80|15.67|34.85|21.48|
> |Transfomrer-3L+Max-Pool|87.39|78.25|**79.81**|15.67|37.72|21.57|
>
> This experiment is performed using the same setup as in table 2 of the main paper.
>
> We observe that the occlusion order performance **marginally improves when increasing the number of
> transformer layers above 1. Worse, it degrades the performance of the model for the depth order
> prediction task.** We believe that this is due to the fact that adding more layers makes the
> optimization procedure harder. It is also better to minimize the number of layers to reduce
> inference time and memory usage. Thus, we opt for a single transformer layer before max-pooling.
>
> > Theoretical advantages of predicting $G$ in one pass.
>
> The main theoretical advantage of predicting $G$ in one pass is that **we achieve $O(1)$ inference
> complexity with respect to the number of instances in the scene**. This claim is empirically
> supported by our runtime/memory experiments in section F figure 7 of the supplementary materials.
>
> We propose an interpretation of why **depth order prediction seems to be easier than occlusion
> prediction in the holistic setting in section E.1 (task difficulty) of the supplementary
> materials.** We kindly refer the reader to the aforementioned section for details. Note that this
> property is unique to the holistic setting and does not emerge in pairwise order prediction
> methods.
>
> ## Limitations
>
> > The paper mentions certain limitations, but it would be beneficial if the method could handle
> > an unlimited number of instances, increasing its practicality and effectiveness in more complex
> > scenarios.
>
> We hope that the previous sections have clarified that our approach can handle easily more than 10
> instances. InstaFormer is, in theory, bounded by the number of queries $N$, **but the table we
> provide in this answer shows that this is not a limitation in practice**. We also **provide failure
> cases of our approach in section G.3 and fig. 9 of the supplementary materials**.
>
> We are planning on **extending this section in the final version of the paper to include more
> failure cases such as the "segmentation failure"** in which the segmentation backbone creates
> multiple masks for the same instance, resulting in multiple nodes for the same object in the
> order graph.
>
> We thank again reviewer NnW7 for their constructive comments. We will integrate them to our
> final version.

---

### Official Review · Reviewer_iz2C · 2025-07-02

**Clarity:** 3
**Significance:** 2
**Originality:** 2
**Rating:** 4
**Confidence:** 3

**Summary:**

The authors introduce a nicer design for order/occlusion prediction by internally combining a segmentation architecture (Mask2Former) and its resulting object queries to predict orderings/occlusion in a full attentional approach. This is easy to train end to end, and can rely more on object-centric context in order to make its predictions. Previous work was less integrated or relies on specific pairwise networks to produce these predictions. The authors show a variety of good baselines (including VLMs, and a more basic Mask2Former approach), while their method generally outperforms existing methods.

**Questions:**

1. More information on how Mask2Order was trained. Is it trained (always) on results from Mask2Former or is this only at inference time?
2. The final prediction is a dot product which is just binary classification. How does this determine the {1, 0, -1} results needed for occlusion ordering? Is it separate MLP for each "class"?

**Ethical Concerns:**

["NO or VERY MINOR ethics concerns only"]

**Final Justification:**

My questions have generally been answered regarding missing ablations and comparisons. I think this paper is still worthy of acceptance, however, I would not necessarily champion it if need be since I think the impact and novelty of the method remains marginal, but this is likely a solid paper in the framework of the task the author's have chosen (i.e., I do not necessarily think the authors can do much to improve this paper in itself).

**Limitations:**

Yes

**Quality:**

2

**Strengths And Weaknesses:**

Strengths:

1. Everything here makes _sense_ and seems superior to previous designs. It's a solid design.
2. This appears to, in turn, show solid results on the ordering/occlusion benchmarks.

Weaknesses:

1. While the authors appear to slight existing methods for requiring masks, the resulting approach of InstaFormer is still lacking in that it requires a full decode of an instance segmentation mask. The main innovation comes the idea that full Transformer attention of mask/image-derived features can be used to perform the pairwise prediction tasks in one shot (e.g. LightGlue, etc). Their ablation shows that it appears to heavily depend on this _explicit_ spatial information which is somewhat disappointing.
2. The authors could spend less prose on comments like "making their real-world inference impractical as the masks might be unavailable to the user." This detracts from the paper since it makes it seem like the idea of using predicted masks instead of the ground-truth masks is a heroic leap of faith.
3. The idea that one can use the dot product here rather than separate classification steps is good, but it seems a little far fetched to call this "holistic". Is this implied from the idea of self attention? Is it from using the dot product for prediction (but this doesn't seem particularly holistic?).

---

> ### Author Rebuttal · Authors · 2025-07-31
>
> # Authors' response to reviewer iz2C
>
> We would first like to thank reviewer iz2C for its detailed and positive feedback.
> We strongly appreciate reviewer iz2C's acknowledgment of the soundness, efficiency and performance
> of our design. We are pleased to hear that they found our baselines comprehensive and benchmark
> convincing. We reply to their considerations as follows:
>
> ## Weaknesses
>
> > InstaFormer requires a full decode of an instance segmentation mask.
>
> We agree that InstaFormer relies on the output obtained from the Mask2Former segmentation
> backbone. **This means that the detection of the instances influences the order prediction of the
> geometrical order module**. We observed cases where instance segmentation failure resulted in order
> prediction mismatch.
>
> Nonetheless, we would like to highlight that **we are not using the masks as raw inputs to our
> geometric order prediction head. Instead, we just use the masks to mask the features of the
> per-pixel decoder.** We believe that this is a substantial nuance that is worth keeping in mind.
> **Our network input's remains in the latent space of the segmentation backbone.**
>
> Moreover, we would like to emphasize that obtaining the segmentation masks as the output of the
> network is something desirable. The segmentation masks are what "ground" the order predictions in
> all cases. Supposing we would not decode the masks, it would be hard to interpret the entries of
> the order matrices since we would not be able to tell which row and column correspond to each
> instance in the image. In the case where the instance label is unique in the scene, one could argue
> that the predicted label would be sufficient to tell the entries apart. However, this solution is
> not robust in the case where multiple instances of the same class are present in the image (which
> is frequent on most natural scenes). **In that case, it is mandatory to obtain the masks from the
> segmentation backbone to be able to interpret the output order graphs properly.**
>
> > About "making real-world inference impractical as the masks might be unavailable to the user".
>
> We hope that this repetition did not interfere with the reading experience in a negative way.
> We believe that mitigating this limitation constitutes an important contribution of our work, yet,
> **we will make sure to only mention it once in the final version.** We agree that this idea does
> is an "heroic leap of faith", yet, we still believe that it constitutes a significant and natural
> evolution in the paradigm of order prediction.
>
> > The use of the dot product rather than separate classification is good but does not seem
> > sufficient to be called "holistic". Is this implied from the idea of self attention?
> > Is it from using the dot product for prediction ?
>
> **The reason why we called our network "holistic" is because it can obtain all the order relations
> in a single forward pass as opposed to previous baselines which only behave at the pair level.**
> This property naturally emerges from formulating the order prediction problem as a full graph
> inference and solving it using a transformer architecture where each token represents a single
> instance. Thus, **reviewer iz2C is right in thinking that this comes from the idea of self
> attention. Since self attention behaves at the token level and since each token has been carefully
> crafted to represent an instance, we naturally draw the order relations for _all_ the object pairs
> in a single transformer pass.** The final dot product only acts as some aggregation operation to
> fall back in the $N \times N$ matrix of $N$ detected instances in the image. One can think of it
> as some object-wise relational compatibility in the latent space.
>
> ## Questions
>
> > More information on how Mask2Order was trained. Is it trained (always) on results from
> > Mask2Former or is this at inference time ?
>
> We thank reviewer iz2C for this question which gives us the opportunity to clarify our setup and
> propose a follow up experiment.
>
> Results reported for Mask2Order in table 1 of the main paper were obtained as follows.
> We trained InstaOrderNet from scratch using the same recipe as in the original InstaOrderNet paper.
> Once training done, we concatenated InstaOrderNet and Mask2Former and conducted evaluation of order
> predictions on the Mask2Former generated masks. We train on InstaOrder and perform inference
> on Mask2Former generated masks. **Results of Mask2Order are at inference time**.
>
> Reviewer iz2C's question gives us the opportunity to train a new version of Mask2Order, which
> uses the **Mask2Former masks as inputs to the order prediction head even during training**. **We
> follow the same training recipe as in the original InstaOrderNets** with a single exception.
> As we empirically found this task harder than order prediction on GT masks, we modernize the
> optimization routine by using AdamW with a learning rate of 0.001 and a cosine decay schedule
> instead of the SGD optimizer with step decay from the original InstaOrderNet paper.
>
> We report the results of this new version of Mask2Order in the following table:
>
> |Method|Precision|Recall|F1|WHDR (distinct)|WHDR (overlap)|WHDR (all)|
> | :- | :-: | :-: | :-: | :-: | :-: | :-: |
> |Mask2Order^od|79.81| 86.86|79.09|12.44|28.44|18.19|
>
> Compared to the original Mask2Order results, we observe a significant accuracy boost for the
> occlusion order prediction task (+1.92 points in F1 score for Mask2Former^od), while the
> depth order performance slightly decreases (+1.1 points in WHDR all for Mask2Former^od).
>
> Yet, these results still all fall short when comparing them to the original InstaOrderNet results,
> highlighting the difficulty of end-to-end order prediction while strengthening the efficiency and
> the importance of our InstaFormer architecture. **Indeed, we observe that the InstaFormer models
> all outperform both frozen _and_ trained Mask2Order models.**
>
> We lacked time to perform the occlusion and depth only experiments in this rebuttal, but will
> add them during the conversation phase (if we have the opportunity) and in the final revision of our manuscript
> along with an analysis. We thank reviewer iz2C for providing this experimental suggestion to us.
>
> > The final prediction is a dot product which is just binary classification. How does this
> > determine the {1, 0, -1} results needed for occlusion ordering? Is it a separate MLP for
> > each "class"?
>
> We encourage the reviewer to follow along Fig. 3 of the main paper to understand the following
> explanations more easily.
>
> **One can think of the final dot product as an object-level relational compatibility**. The
> representations of the object queries features $Q^\*$ and object descriptors $D^\*$ are both
> $N \times C$ and $N \times C$. Thus, $G = Q^\*D^{\*T}$ results in a $N \times N$ matrix which is
> sent to a final MLP (MLP$_{\omega}$ in figure 3 of the main paper. **Note that the final MLP is
> task dependent, meaning there are 1 MLP for occlusion and 1 MLP for depth**).
>
> **The output dimension of the final MLP is either 2 for the occlusion task or 3 for the depth
> ordering task.** Indeed, the occlusion task requires to predict whether an object is in
> occluding/hiding, with label 1, or has no occlusion relation with another object, with label 0.
> The depth order prediction task requires to predict whether an object is in front of another,
> with label 1, or behind with label 0. There is a third label, label 2, which corresponds to
> objects sharing the same depth range.
>
> We manually remove the diagonal of the matrix since it does not make sense for an object to
> occlude itself or be in front/behind itself and compute the loss on all the remaining elements of
> the matrix. **Thus, we confirm to reviewer iz2C that there are no separate MLPs for each class.**
> **The final MLP treats all entries in the matrix uniformly and its goal is simply to output proper
> dimensions for each of the tasks.**
>
> **We use BCE for the occlusion order prediction task and CE for the depth order prediction task.**
> At inference, we argmax the predictions on both the occlusion and depth order logits and manually
> fill the diagonal with -1 (this value is arbitrary).
>
> If more questions arise, we kindly refer you to our answer to reviewer i3cT where we provide
> explanations to other details of the tasks. If your questions remain, we would be pleased to
> answer them during the discussion phase.

---

> > ### Author Response · Authors · 2025-08-06
> >
> > Dear reviewer iz2C,
> >
> > We hope this message finds you well. We are writing to kindly follow up on our manuscript rebuttal.
> > We would be grateful for any feedback at your convenience. We hope that our rebuttal have
> > addressed questions and concerns you may have had. We would be more than happy to provide any
> > additional information or clarifications if needed.
> >
> > Thank you very much for your time and consideration.
> >
> > Best regards,
> >
> > > Authors of Submission9867

---

> > ### Comment · Reviewer_iz2C · 2025-08-06
> > **Response**
> >
> > I thank the authors for their response. My questions have generally been answered. I think this paper is still worthy of acceptance, however, I would not necessarily champion it if need be since I think the impact and novelty of the method remains marginal, but this is likely a solid paper in the framework of the task the author's have chosen (i.e., I do not necessarily think the authors can do much to improve this paper in itself).

---

> > > ### Author Response · Authors · 2025-08-07
> > >
> > > Dear reviewer iz2C,
> > >
> > > We are glad we could answer your questions properly. We are delighted to hear your positive feedback and the fact that the state of the paper seems satisfactory to you. The precious feedback from your review helped us to substantially polish our manuscript. We deeply thank you for this.
> > >
> > > > Authors of Submission9867

---

### Official Review · Reviewer_SiaZ · 2025-07-08

**Clarity:** 3
**Significance:** 2
**Originality:** 3
**Rating:** 4
**Confidence:** 3

**Summary:**

The authors present a method for predicting object order (occlusion relations and depth ordering). The nice new aspect of the method is that it generates order in a single forward pass by modeling the interaction between object queries and latent mask descriptors. The experimental results are comprehensive, by comparing the method to past works, including three parallel works. Results are promising.

**Questions:**

- Motivation is a bit weak. Highlighting an important use case could make a reader more excited about the area of work.

**Ethical Concerns:**

["NO or VERY MINOR ethics concerns only"]

**Limitations:**

yes

**Quality:**

3

**Strengths And Weaknesses:**

Strengths:
- Modeling order in a single forward pass with good results.
- Motivating why order is important by linking it with the VLM which improves performance in depth based reasoning.
- Clear write up
- Comprehensive experiments

Weaknesses:
- The related work section could benefit from the authors reviewing some older works on the subject. For example, "Monocular Object Instance Segmentation and Depth Ordering with CNNs" by Zhang et al plus references within could be a good reference for older works.
- Motivating why depth ordering is important could be done more convincingly. I'm still not sure where someone would actually use this method in practice.

---

> ### Author Rebuttal · Authors · 2025-07-31
>
> # Authors' response to reviewer SiaZ
>
> We deeply appreciate reviewer SiaZ's thoughtful comments and suggestions. We are glad they found
> our method interesting, our benchmark comprehensive and our writing clear. We address each of
> their comments below:
>
> ## Weaknesses
>
> > The related section could benefit from the authors reviewing older works on the subject.
> > For example, "Monocular Object Instance Segmentation and Depth Ordering with CNNs" by Zhang
> > et al. and references therein.
>
> We thank reviewer SiaZ for pointing this relevant work to us. We agree that this paper falls in
> the range of our work by proposing a pioneering approach in the field of depth order prediction.
> While the suggested work did not provide open-source implementation of their method, **we plan
> on citing it in the related work section of the final version of our paper.**
>
> Thanks to reviewer SiaZ's suggestion, we also found the paper "A Learning-Based Framework for
> Depth Ordering" by Z. Jia et al. to be a relevant depth order art and will also cite
> it in the final version of our paper.
>
> We will continue to look for other relevant works and make sure to include them.
>
> > Motivating why depth order is important could be done more convincingly. I'm still not sure why
> > someone would actually use this method in practice.
>
> We thank the reviewer for offering us the opportunity to clarify the motivations behind predicting
> depth order relations in a scene. We currently think that there are several compelling areas in
> which depth order relations are crucial and where our method can be applied effectively:
>
> 1. **Real world applications**
>    - **Autonomous driving**: understanding the depth order of objects in a scene is crucial for
>      safe navigation and obstacle avoidance.
>    - **Augmented reality (AR)**: depth ordering is essential for rendering virtual objects in a
>      way that they appear naturally integrated into the real world.
>    - **Positional verification in sports**: in sports, depth order awareness can become crucial to
>      assess if a rule has been violated. We give a two examples here:
>      - In athletics: depth orders can determine the finish order of runners (as illustrated
>        in Fig. 1 of the main paper).
>      - In football: depth orders can be used to determine if a player is offside or not.
> 2. **Vision language models (VLMs)**
>    - We show that predicting order graphs before VLM inference can improve the spatial
>      understanding of VLMs in a zero-shot manner (Fig. 1 of the main paper). This is particularly
>      useful since VLMs often struggle with spatial reasoning, while fine-tuning them is usually
>      expensive in terms of data and compute resources.
> 3. **Robotics**
>    - Recently, VLMs have been used in robotics in the form of perception modules.
>      Thus, obtaining fine-grained spatial relations of objects in the surrounding world becomes
>      crucial for grounded and safe interactions between the robot and its environment. In some
>      cases the description of an object to pick up or manipulate are given by its location in space
>      (e.g. "the object behind the red box"), which can be poorly interpreted by VLMs. As shown in
>      Fig. 1 of the main paper, providing depth order relations can help the robot to identify the
>      object to be manipulated accurately.
> 4. **Image editing**:
>    - Recent works such as "LooseControl: Lifting ControlNet for Generalized Depth Conditioning"
>      S. F. Bhat et al., or "Build-A-Scene: Interactive 3D Layout Control for Diffusion-based
>      Image Generation" by A. Eldesokey & P. Wonka, use monocular depth estimation as a mean
>      to edit images efficiently. Although this requires further research, we believe that our
>      method could be used to build similar models without relying on a full depth map of the scene.
>      Generally speaking, the field of layered image editing, where each object is considered as a
>      different plane on the 3D space, is a promising area where our approach has chances to be
>      applied effectively.
>
> As suggested by the reviewer, the paper "Monocular Object Instance Segmentation and Depth
> Ordering with CNNs" also provides examples of cases where our approach could be used in practice:
> driver assistance, image captioning with spatial arrangements, Q&A retrieval systems, and 3D scene
> description.
>
> **We will rewrite some part the introduction to include some of these examples and motivate
> our work more effectively.**
>
> We hope that these examples clarify the motivations behind our work and its potential applications.
> We thank reviewer SiaZ again for giving us the opportunity to clarify these important points.

---

> > ### Author Response · Authors · 2025-08-06
> >
> > Dear reviewer SiaZ,
> >
> > We hope this message finds you well. We are writing to kindly follow up on our manuscript rebuttal.
> > We would be grateful for any feedback at your convenience. We hope that our rebuttal have
> > addressed questions and concerns you may have had. We would be more than happy to provide any
> > additional information or clarifications if needed.
> >
> > Thank you very much for your time and consideration.
> >
> > Best regards,
> >
> > > Authors of Submission9867

---

> > ### Author Response · Authors · 2025-08-09
> >
> > # Authors' comprehensive summary
> >
> > Dear reviewer SiaZ, we hope this message finds you well. We are thankful for your detailed
> > review and for your comments on our manuscript. We believe that this helped us refine it
> > significantly the motivations behind it.
> >
> > We would like to quickly summarize the elements we addressed in our rebuttal.
> >
> > > Proposition to add related works
> >
> > We thank reviewer SiaZ again for this suggestion which help us frame our work in the larger
> > context of the literature. We added the proposed related works to our manuscript.
> >
> > > Motivation behind the order prediction task
> >
> > We proposed multiple use cases in which determining the depth order of objects is critical
> > (real world applications, VLMs, robotics, image editing). We hope this short list will
> > help the reviewer to better understand the importance of this task.
> >
> > We once again thank you for your time and effort in reviewing our submission. We hope that
> > the changes we made in the manuscript will address your concerns and that you will find our
> > motivations strong.
> >
> > We are always open to further discussions and clarifications before the deadline of the
> > discussion phase.
> >
> > We would once again express our gratitude towards the time you spend on reviewing NeurIPS
> > submissions.
> >
> > Best regards,
> >
> > > Authors of Submission9867

---

### Comment · Area_Chair_U6xq · 2025-08-05
**Please participate in reviewer-author discussion**

Dear Reviewers,

Thanks again for serving for NeurIPS, please read the rebuttal and discuss with the authors if you have any follow-up questions. The deadline of author-reviewer discussion is Aug 8, 11.59pm AoE.

Thanks,

AC

---

### Comment · Area_Chair_U6xq · 2025-08-06
**Please participate in discussion**

Dear Reviewers,

The reviewer-author discussion ends in two days ( Aug 8, 11.59pm AoE. ). Please read the rebuttal and follow up with the authors if you have further questions. Your timely response is highly appreciated.

Thanks

AC

---

### Decision · Program_Chairs · 2025-09-17

**Decision:**

Accept (poster)

**Comment:**

This paper proposes a feed-forward framework for predicting object ordering in single-view images. The central component is a transformer-based model that takes an image as input and predicts two relational matrices, capturing pairwise occlusion and depth ordering among objects. The model is trained end-to-end and evaluated against both heuristic and learning-based baselines.

All reviewers consistently recommend acceptance. They find the task to be well-motivated and of broad interest, and agree that the proposed solution is both effective and practically relevant. After carefully considering the paper, the reviews, and the rebuttal, AC concurs with the reviewers’ assessment: the paper addresses an important problem and demonstrates state-of-the-art performance. I therefore recommend acceptance.

As emphasized by the reviewers and AC, the authors are encouraged to incorporate the clarifications, discussions, and improvements raised during the rebuttal into the final version.